# Learning Molecular Symmetry Breaking via Symmetry-adapted Neural Networks

## Abstract

E(3)-equivariant neural networks have achieved remarkable performance in molecular modeling. However, the equivariance constraint limits the model's effectiveness in learning tasks involving symmetry breaking, particularly those that violate the celebrated Curie principle. Relaxing the equivariance constraint is essential for addressing these challenges. In this paper, we explore the intricate symmetry relationships between an object and its spontaneously symmetry-broken outcomes. We introduce a relaxed equivariance based on the molecule's inherent symmetries. Additionally, we develop SANN – a symmetry-adapted neural network architecture that learns symmetry breaking through equivalence classes of atoms. SANN decomposes the molecular point cloud into sets of symmetry-equivalent atoms and performs message-passing both within and across these classes. We demonstrate the advantages of our method over invariant and equivariant models through synthetic tasks and show that SANN effectively learns both equivariance and symmetry breaking in various benchmark molecular modeling tasks.

## 1 Introduction

Molecular modeling – including tasks such as material and drug design, protein structure prediction, and analyzing protein structure-function relationships – is one of the foremost scientific challenges (Moult, 2005; Lavecchia, 2019; Tibbitt et al., 2015). Recently, significant strides have been made in machine learning (ML)-assisted molecular modeling (Duvenaud et al., 2015; Gilmer et al., 2017; Jumper et al., 2021; Merchant et al., 2023; Wang et al., 2023a; Zhang et al., 2023). Equivariant models are especially promising – they ensure their output transforms appropriately when the input data undergoes a transformation, such as a rotation or translation.

Euclidean-equivariant graph neural networks (GNNs), especially, the E(n)-equivariant GNN (EGNN) (Satorras et al., 2021), have been the backbone of many molecular predictive and generative models. Indeed, these equivariant models have achieved appealing results in molecular properties prediction (Schütt et al., 2021; Zhang et al., 2024), molecular dynamics simulation (Batzner et al., 2022), crystal design (Xie et al., 2022; Luo et al., 2023), protein-docking (Ganea et al., 2022; Yim et al., 2024), and protein design (Lin & AlQuraishi, 2023; Watson et al., 2023). By enforcing the principle of Euclidean equivariance, these models ensure their predictions are transformed equivariantly to rotations, reflections, and

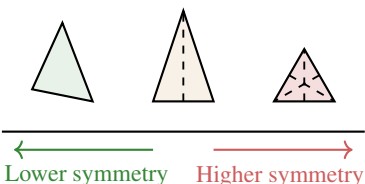

Lower symmetry    Higher symmetry

Figure 1: An illustration of a triangle with different symmetries: An equilateral triangle (marked in red) exhibits the highest symmetry, while an isosceles triangle (marked in yellow) has lower symmetry, and a scalene triangle (marked in green) has no symmetry.

translations of their input data. This equivariance property serves as a crucial inductive bias, guiding the learning process toward models that are accurate and generalizable to unseen data.

Despite being a cornerstone of many successful molecular modeling applications, the adherence of Euclidean equivariance to the *Curie principle* (see e.g., (Curie, 1894; Smidt et al., 2021)) can limit equivariant models' capability to capture certain physical phenomena. This principle asserts that the output of an equivariant function cannot have a lower symmetry[1] than the input. However, many physical processes described by order-disorder and structural transitions (Kivelson et al., 2024),

---

[1]Symmetry in a system is characterized by the group actions that do not change its configuration.

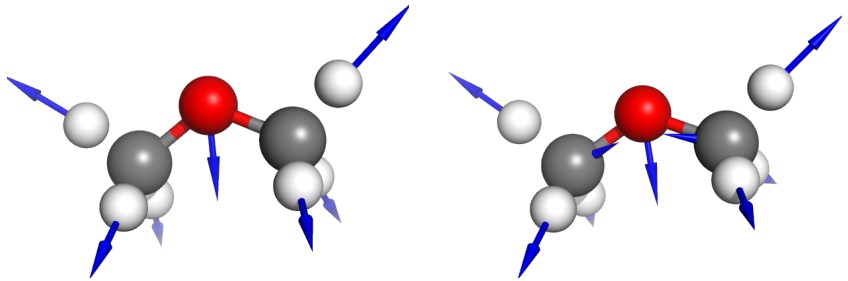

Figure 2: Symmetry breaking occurs in the QM7-X dataset. These molecules are conformers of methoxy-methane ($CH_3OCH_3$). The Hirshfeld dipole moment vectors on the left respect molecular symmetries, unlike those on the right. This is apparent when viewing the contribution from the carbons (gray atoms), where on the left, they contribute along the oxygen (red atom) bond directions – but on the right, they deviate significantly.

such as protein self-assembly (La Rosa et al., 2020) and cell polarization (Miller et al., 2022) exhibit *spontaneous symmetry breaking* (SSB) (Castellani & Dardashti, 2021; Wang et al., 2024). During SSB, a system transitions from a state with higher symmetry to another state with lower symmetry. The concept of symmetry breaking (SB) is illustrated in Fig. 1: an equilateral triangle (marked in red) exhibits the highest symmetry, while an isosceles triangle (marked in yellow) has lower symmetry, and a scalene triangle (marked in green) has no symmetry. Though equivariant models can learn the transformation from an isosceles triangle to an equilateral triangle (an increase in symmetry), they cannot learn SB transformations from an isosceles triangle to a scalene triangle (a decrease in symmetry).

While SB has been explored in various ML contexts, to the best of our knowledge, none have addressed SB in molecules. Figure 2 shows an example of such SB in conformers of methoxymethane ($CH_3OCH_3$) – an organic compound. Specifically, the blue arrows represent the Hirshfeld dipole moment vectors of each atom. On the left, these vectors align with molecular symmetries, particularly the blue arrows from the carbon atoms (gray), which symmetrically contribute along the bond directions of the oxygen atom (red). In contrast, on the right, the vectors deviate significantly, pointing away from the oxygen atoms and breaking one of the molecule's planes of reflectional symmetry.

It is worth noting that SSB often involves ambiguity, where multiple equivalent outcomes are possible, making the task non-functional[2]. Figure 3 shows a classic SSB task – transforming a square into a rectangle (Wang et al., 2023b; 2024). A square has higher symmetry than a rectangle, so this transformation breaks some symmetry of the square. Consider the transformation that stretches the square horizontally or vertically, resulting in two different rectangles, A and B, respectively. Despite their different appearances, these rectangles should be considered equivalent from the perspective of the input square; this is because the two stretching directions (marked as internal gray arrows) are indistinguishable from the input's (square) perspective. So, from the square's point of view, both stretching directions

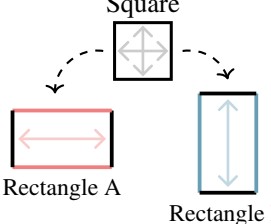

Figure 3: An example of SSB, transforming a square into Rectangle A or B. The gray arrows represent indistinguishable stretching directions of the square, showing A and B are equivalent outputs.

are "identical" ways to become a rectangle. The non-functional nature of SB hinders the models' capability to learn and generalize. We provide a detailed discussion on SSB in Section 3.

## 1.1 OUR CONTRIBUTION

While SB has been explored in various ML contexts, its application to ML-assisted molecular modeling problems remains largely unexplored. Moreover, existing techniques for molecular structure representation often focus on equivariance, which is not applicable when SB occurs. These techniques also fail to incorporate the inherent molecular symmetries, limiting their ability to effectively capture SB phenomena, especially the ambiguity of SB outcomes. In this paper, we aim to address these limitations by introducing the first explicit construction of equivariant neural networks that

---

[2]Here, non-functional means one input can correspond to multiple different outputs.

enable learning SB using GNNs that directly leverage molecular inherent symmetries. In particular, we summarize our main contributions in this paper as follows:

1. We propose to achieve relaxed equivariance using the canonicalization of functions leveraging molecule's inherent symmetries. The relaxed equivariance efficiently replaces the traditional equivariance constraint to accommodate SSB.

2. We analyze molecular symmetries by examining the symmetries of its constituent atoms. Based on these insights, we propose performing message passing within and across sets of symmetry-equivalent classes. This approach is implemented in our symmetry-adapted neural network (SANN) architecture, which learns features under varying symmetry conditions, including invariant, equivariant, and symmetry-broken features.

3. We develop an algorithm to explicitly construct the stabilizers for molecular structures enabling the broad use of a SB measure. This algorithm utilizes successive symmetry checking to compute and construct the stabilizers for a given point cloud.

We substantiate the effectiveness of the proposed SANN architecture using both synthetic tasks and molecular modeling benchmarks.

## 1.2 Organization

In Section 2, we provide a review of some necessary background materials. We present the proposed relaxed equivariance through molecular symmetries in Section 3. We analyze molecular symmetries by examining the symmetries of its constituent atoms and introduce our proposed SANN architecture in Sections 4 and 5, respectively. In Section 6, we verify the efficacy of the proposed algorithms using a few molecular modeling benchmarks. We discuss some additional related works in 7. Missing proofs and additional details are provided in the appendix.

## 2 Background and Some Related Works

In this section, we aim to provide a brief review of some crucial concepts that serve as the pillar of our work. In particular, we recall some concepts of molecular symmetries, equivariance relation and the Curie principle, and atomic and molecular orbitals.

**Molecular representation and symmetries.** A molecule can be represented as a set of tuples, denoted as $\mathcal{M} = \{(\boldsymbol{x}_i, a_i)\}$, where each tuple $(\boldsymbol{x}_i, a_i)$ represents an atom with the atomic number $a_i$ located at $\boldsymbol{x}_i \in \mathbb{R}^3$. A spatial transformation $g$ (e.g., rotation) on the molecule is defined as $g \cdot \mathcal{M} := \{(g \cdot \boldsymbol{x}_i, a_i)\}$. The (inherent) symmetry of a molecule $\mathcal{M} = \{(\boldsymbol{x}_i, a_i)\}$ is characterized by its *point group*, denoted by $\mathrm{Sym}\,(\mathcal{M})$, consisting of transformations in $\mathrm{O}(3)$ that preserve both atom types and atoms' spatial arrangement. Formally, $g \in \mathrm{Sym}\,(\mathcal{M}) \subseteq \mathrm{O}(3)$ if and only if applying $g$ to molecule $\mathcal{M}$ results in the same geometric configuration of atoms, that is, $g \cdot \mathcal{M} = \mathcal{M}$.

The point group $\mathrm{Sym}\,(\mathcal{M})$ induces an equivalent relation: $(\boldsymbol{x}_i, a_i) \sim (\boldsymbol{x}_j, a_j)$ if $(\boldsymbol{x}_j, a_j) = (g \cdot \boldsymbol{x}_i, a_i)$ for some $g \in \mathrm{Sym}\,(\mathcal{M})$. This equivalence relation partitions $\mathcal{M}$ into mutually exclusive subsets, called *equivalence classes*. We denote the equivalence class that consists of atoms equivalent to $(\boldsymbol{x}_i, a_i)$ as $[(\boldsymbol{x}_i, a_i)] = \{(g \cdot \boldsymbol{x}_i, a_i) \mid g \in \mathrm{Sym}\,(\mathcal{M})\}$. The *quotient set* – contains all equivalence classes – is then defined as $\mathcal{M}/\mathrm{Sym}\,(\mathcal{M}) := \{[(\boldsymbol{x}_i, a_i)] \mid (\boldsymbol{x}_i, a_i) \in \mathcal{M}\}$. Moreover, notice that $\mathcal{M}$ is a disjoint union of all equivalence classes in $\mathcal{M}/\mathrm{Sym}\,(\mathcal{M})$, i.e., $\mathcal{M} = \bigsqcup_{[(\boldsymbol{x}_i, a_i)] \in \mathcal{M}/\mathrm{Sym}(\mathcal{M})} [(\boldsymbol{x}_i, a_i)]$.

**Equivariance and Curie principle.** The *equivariance* of a function $f : X \to Y$ with respect to the action of a group $G$ is defined as follows:

$$f(g \cdot x) = g \cdot f(x), \quad \text{for all } g \in G.$$

In particular, if the action on $Y$ is trivial, this reduces to $f(g \cdot x) = f(x)$, which defines an *invariant* function. Equivariant functions obey the Curie principle, i.e., the symmetries of the input must be contained in the symmetries of the output. In other words, we have the following relationship:

$$\mathrm{Sym}\,(x) \subseteq \mathrm{Sym}\,(f(x)).$$

**Atomic orbitals and molecular orbitals.** Here, we provide a brief review of atomic orbitals and molecular orbitals, we refer readers to (Tsaparlis, 1997; Ching & Rulis, 2012) for details. Atomic

orbitals describe the probability distribution of finding an electron around an atom's nucleus, and these atomic orbitals are often approximated as linear combinations of basis functions, such as Slater-type orbitals (STOs) or Gaussian-type orbitals (GTOs). These basis functions are typically expressed in terms of radial and angular components, following a radial-angular decomposition

$$\Phi(\boldsymbol{x}) = R_l(r)Y_m^l(\theta, \phi), \tag{1}$$

where $\boldsymbol{x} = (r, \theta, \phi)$ are spherical coordinates and $Y_m^l$ is a spherical harmonics and $R_l(r)$ is the radial part, $l$ and $m$ are the orbital angular momentum and its $z$ component. To better understand the molecular structure, bonding, and reactivity of a molecule $\mathcal{M} = \{(\boldsymbol{x}_i, \boldsymbol{a}_i)\}$, a well-known technique called *linear combination of atomic orbitals* (LCAO) is often used. In this method, the basis of molecular orbitals is expressed as a linear combination of atomic orbitals, which helps us to describe whether atomic orbitals combine to form bonds or antibonds in a molecule. The expression for the $k^{th}$ basis of the molecular orbital is written as follows:

$$\Phi_k^{\mathcal{M}}(\boldsymbol{x}) = \sum_{(\boldsymbol{x}_j, a_j) \in \mathcal{M}} c_{kj}\Phi_{(\boldsymbol{x}_j, a_j)}(\boldsymbol{x}), \tag{2}$$

where $\Phi_{(\boldsymbol{x}_j, a_j)}(\boldsymbol{x}) \coloneqq \Phi(\boldsymbol{x} - \boldsymbol{x}_j)$ denotes the atomic orbital centered at the nucleus of atom $(\boldsymbol{x}_i, a_i)$.

## 3 SPONTANEOUS SYMMETRY BREAKING AND RELAXED EQUIVARIANCE

In this section, we begin by reviewing the concept of SSB. Then we formally characterize the inherent ambiguity associated with multiple equivalent outcomes of SSB; in particular, we employ the recently proposed loss function by Xie & Smidt (2024) to measure this inherent ambiguity. We further delve into the notion of *relaxed equivariance* through *canonicalization* introduced in (Kaba et al., 2023; Baker et al., 2024). This approach allows for a more flexible utilization of symmetries, especially in situations where strict equivariance may be overly restrictive. We explore how relaxed equivariance can enhance the learning of SB in terms of data efficiency.

### 3.1 SPONTANEOUS SYMMETRY BREAKING

While the Curie principle applies to many physical systems, there are real-world phenomena that do not strictly adhere to it. SSB is a notable exception (Beekman et al., 2019; Castellani & Dardashti, 2021), where the input exhibits a higher symmetry than the output. In SSB, the input can *spontaneously* lose its symmetry and transition to one of several equivalent outputs, denoted by $\mathrm{SSB}(x) \coloneqq \{g \cdot y \mid g \in \mathrm{Sym}(x)\}$, with lower symmetry. These outputs differ only by the symmetries of the input $x$, meaning they share the same type of reduced symmetries, with their symmetry groups related by conjugation. To be more specific, for each possible output $y_i \in \{g \cdot y \mid g \in \mathrm{Sym}(x)\}$, we have

$$\mathrm{Sym}(y_i) \subsetneq \mathrm{Sym}(x), \text{ indicating that } y_i \text{ has lower symmetry than } x.$$

Additionally, since two valid outputs $y_i, y_j \in \{g \cdot y \mid g \in \mathrm{Sym}(x)\}$ differ only by a symmetry operation of $x$, say $y_i = g_{ij} \cdot y_j$ for some $g_{ij} \in \mathrm{Sym}(x)$, it follows that $\mathrm{Sym}(y_j) = g_{ij}^{-1}\mathrm{Sym}(y_i)g_{ij}$, implying that these outputs shared the same reduced symmetry. An example of SSB is the transformation of a square into rectangles, as illustrated in Fig. 3. We can observe that the two rectangles differ only by the rotational symmetry of the square and share the same type of symmetries. Moreover, these two rectangles are equivalent SSB outcomes of the square, as the two different stretching directions are indistinguishable from the perspective of the square itself.

### 3.2 SYMMETRY BREAKING MEASURE

Mathematically, SSB can be expressed as a set-valued function, mapping $x$ to the set of equivalent outputs $\mathrm{SSB}(x)$. However, it's important to note that while an input may transition to different outcomes at different times, it can only occupy one output state at a specific moment. Following (Xie & Smidt, 2024), we define a single-valued function $f : X \to Y$ as *learning the SSB of $x \in X$* if $f(x)$ lies in the set of equivalent outputs $\mathrm{SSB}(x)$, i.e., $f(x) \in \mathrm{SSB}(x)$.

Consider a learnable function $f_\theta$. Suppose the training data includes only one possible outcome of SSB for a given input $x$, says $y \in \mathrm{SSB}(x)$. If the function predicts a different valid output $f_\theta(x) = y' \in \mathrm{SSB}(x)$, traditional loss functions like mean squared error (MSE) or cross-entropy will report a non-zero value, even though both $y$ and $y'$ are valid. Moreover, if the training data contains multiple equivalent outcomes $y, y' \in \mathrm{SSB}(x)$ for $x$, traditional loss functions might confuse the model, leading it to associate the incorrect output with the input. To address these challenges

and ensure accurate loss measurement, we employ the symmetry breaking measure (SBM) proposed by Xie & Smidt (2024), which is given by

$$\text{SBM}\left(f, \{(x_k, y_k)\}_{k=1}^K\right) := \frac{1}{K} \sum_{k=1}^K \min_{g \in \text{Sym}(x_k)} m(f_\theta(x_k), g \cdot y_k), \tag{3}$$

where $\{(x_k, y_k)\}_{k=1}^K$ is the set of observed SSB data and $m(y', y) \geq 0$ is a suitable metric that is 0 if and only if $y' = y$. Note that all the possible outcomes of the SSB for $x_k$ must lie in $\{g \cdot y_k \mid \text{Sym}(x_k)\} = \text{SSB}(x_k)$. Therefore, the function $f_\theta$ learns the SSB for $x_k$ if and only if $\min_{g \in \text{Sym}(x_k)} \|f_\theta(x_k) - g \cdot y_k\| = 0$. In conclusion, the SBM calculates the minimum across the distance between the predicted output $f_\theta(x_k)$ and any equivalent output $g \cdot y_k$, effectively addressing the ambiguity associated with SB.

*Remark* 3.1. When the input $x_k$ does not exhibit any symmetry, i.e., $\text{Sym}(x_k) = \{e\}$ where $e$ is the identity element in $\text{O}(3)$, or when the relationship between $x_k$ and $y_k$ is equivariant, meaning $\text{Sym}(x_k) \subseteq \text{Sym}(y_k)$, the measure $\min_{g \in \text{Sym}(x_k)} m(f_\theta(x_k), g \cdot y_k)$ on the data pair $(x_k, y_k)$ simplifies to the standard loss $m(f_\theta(x_k), y_k)$.

*Remark* 3.2. The work of (Xie & Smidt, 2024) is limited by prior knowledge of the symmetries in the input and the output. The canonicalization algorithm introduced in (Baker et al., 2024) can be improved to provide an explicit method for determining the symmetries and generating $\{g \cdot y_k \mid \text{Sym}(x_k)\}$ for any point cloud. A detailed description of this algorithm is provided in Appendix B. We also provide analysis for the computational complexity, and an ablation study of this measure with and without SANN architectures, in Section 5.

### 3.3 RELAXED EQUIVARIANCE AND CANONICALIZATION

As discussed in Section 1, strict equivariance, which adheres to Curie's principle, can be too restrictive for SSB tasks (see Fig. 3). To maintain the data efficiency advantages of equivariance while accommodating the inherent ambiguity of SSB outcomes, we propose relaxing the strict equivariance requirement. Consider the scenario depicted in Fig. 4. Given a square $x$ (in the upper left corner) and its rotated version $g \cdot x$ where $g$ is some rotation. Both undergo SSB characterized by a function $f$, producing rectangles. However, the resulting rectangles do not

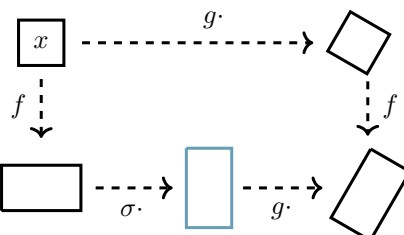

Figure 4: Illustration of relaxed equivariance.

simply differ by $g$ because there are multiple valid outputs for SB (see Fig 3). If we examine another possible output $\sigma \cdot f(x)$ for the input square $x$, we observe that it only differs from $f(g \cdot x)$ by $g$. Therefore, we conclude that $f(g \cdot x) = g \cdot \sigma \cdot f(x)$, where $\sigma$ accounts for the SB ambiguity. Based on this observation, we introduce the following notion of relaxed equivariance in (Kaba et al., 2023), which is well-suited for modeling SSB scenarios and accurately characterizes the weaker equivariant relationships involved:

**Definition 3.3.** Let $f : X \to Y$ be a function between spaces $X, Y$ where a group action of $\text{O}(3)$ is defined on both spaces. We say a function is *relaxed equivariant* with respect to $\text{O}(3)$ if, for any $x \in X$ and $g \in \text{O}(3)$, there is an element $\sigma \in \text{Sym}(x)$ such that

$$f(g \cdot x) = g \cdot \sigma \cdot f(x). \tag{4}$$

or equivalently, $g^{-1} \cdot f(g \cdot x)$ differs from $f(x)$ only by $\sigma$ – a symmetry operation of $x$.

*Remark* 3.4. Notice that $g^{-1} \cdot f(g \cdot x)$ and $f(x)$ are the same for a strictly equivariant function.

To achieve relaxed equivariance, we propose canonicalizing the learnable model as presented in (Baker et al., 2024). Canonicalization allows the model to focus on learning from a single representative instance of each data class and automatically generalizes to transformed input through relaxed equivariance. We summarize the notion of canonicalization and its properties in the following:

**Definition 3.5.** Let $X$ be the set where the group $\text{O}(3)$ acts. A *frame* on $X$ is a relaxed equivariant function from $X$ to $\text{O}(3)$, which induces an $\text{O}(3)$-invariant function $c : X \to X$, called a *canonicalization*, by $c(x) := \mathcal{F}(x)^{-1} \cdot x$. The output of $c$ at $x$, denoted by $c(x)$, is referred to as the canonical representative of $x$. For any function $f : X \to Y$ between spaces where $\text{O}(3)$ acts, we define the *canonicalization* of $f$ through the frame $\mathcal{F}$ as follows:

$$f_{\text{canonical}}(x) := \mathcal{F}(x) \cdot f(\mathcal{F}(x)^{-1} \cdot x). \tag{5}$$

**Proposition 3.6.** *Let $\mathcal{F} : X \to \mathrm{O}(3)$ be a frame on $X$. Then for any function $f : X \to Y$ between spaces with group actions, the canonicalization $f_{\mathrm{canonical}}$ is relaxed equivariant to $\mathrm{O}(3)$. In particular, $f_{\mathrm{canonical}}$ is equivariant at $x \in X$ if $f$ is equivariant on $x$.*

As we can see from Definition 3.5, the canonicalization of a function is only evaluated on the canonical representatives in the image $c(X)$. Importantly, canonicalization does not compromise the universality of the learning framework (Baker et al., 2024). Hence, it suffices to consider learning SB for these canonical representatives. The prediction will then be automatically generalized to the entire c(X) due to the relaxed equivariance proved in Proposition 3.6.

## 4 SYMMETRY-ADAPTED LINEAR COMBINATIONS

To effectively learn SB, we explore the intricate composition of symmetries within the data. We begin by formalizing the relationship between the symmetry of the entire molecule $\mathcal{M}$ and the symmetries of its individual equivalence classes in $\mathcal{M}/\mathrm{Sym}\,(\mathcal{M})$:

**Proposition 4.1.** *The point group of a molecule $\mathcal{M}$ is the intersection of the point groups of its equivalence classes. In particular, we have:*

$$\mathrm{Sym}\,(\mathcal{M}) = \bigcap_{[(\boldsymbol{x}_i, a_i)] \in \mathcal{M}/\mathrm{Sym}(\mathcal{M})} \mathrm{Sym}\,([(\boldsymbol{x}_i, a_i)]). \tag{6}$$

In simpler terms, the symmetry of the entire molecule is determined by the symmetries present within the sets of its equivalent atoms. This implies that when SB occurs, it must manifest in at least one of these equivalence classes. We summarize this implication below:

**Corollary 4.2.** *If a symmetry breaking $f$ occurs for a molecule $\mathcal{M}$, resulting in $f(\mathcal{M})$, then any broken symmetry must occur in at least one of the equivalence classes in $f(\mathcal{M})$.*

Based on the insights from Proposition 4.1 and Corollary 4.2, especially when the molecule exhibits symmetry, we propose a new framework that first learns features at the level of equivalence classes, rather than directly from neighboring nodes in the graph. This approach effectively captures SB, as it is directly tied to symmetry breaking within a class of equivalent atoms. We then aggregate the learned features across equivalence classes to retain the overall symmetry of the molecule, as inspired by the intersection described in Proposition 4.1. This framework enables a more targeted and efficient detection of symmetry breaking within the molecular structure. In particular, we refine the linear combination expression in equation 2 to capture SB in molecules. For a class of equivalent atoms $[(\boldsymbol{x}_i, a_i)] \in \mathcal{M}/\mathrm{Sym}\,(\mathcal{M})$, we consider the linear combinations of their orbitals

$$\Phi_k^{[(\boldsymbol{x}_i, a_i)]}(\boldsymbol{x}) = \sum_{(\boldsymbol{x}_j, a_j) \in [(\boldsymbol{x}_i, a_i)]} c_{kj}(\boldsymbol{x}_j, a_j) \Phi_{(\boldsymbol{x}_j, a_j)}(\boldsymbol{x}), \tag{7}$$

where the sum is taken over all atoms in the symmetry-equivalence class. By combining these linear combinations across all equivalence classes, we can express the basis of molecular orbitals as:

$$\Phi_k^{\mathcal{M}}(\boldsymbol{x}) = \sum_{[(\boldsymbol{x}_i, a_i)] \in \mathcal{M}/\mathrm{Sym}(\mathcal{M})} \Phi_k^{[(\boldsymbol{x}_i, a_i)]}(\boldsymbol{x}). \tag{8}$$

Combining these two summations, we have:

$$\Phi_k^{\mathcal{M}}(\boldsymbol{x}) = \sum_{[(\boldsymbol{x}_i, a_i)] \in \mathcal{M}/\mathrm{Sym}(\mathcal{M})} \sum_{(\boldsymbol{x}_j, a_j) \in [(\boldsymbol{x}_i, a_i)]} c_{kj}(\boldsymbol{x}_j, a_j) \Phi_{(\boldsymbol{x}_j, a_j)}(\boldsymbol{x}). \tag{9}$$

Note that the decomposition of molecular orbitals proposed in equation 9 closely resembles the real symmetry-adapted linear combinations (SALCs) used in quantum chemistry. However, in our approach, the coefficients $c_{kj}(\boldsymbol{x}_j, a_j)$ are learned rather than determined by group representations. For a more in-depth discussion of SALCs, please refer to (Kim, 1999). For convenience, we will refer to the proposed decomposition in equation 9 by the same name.

This decomposition aligns with the insights from Proposition 4.1 and Corollary 4.2, allowing us to infer the symmetries or SB of the entire molecule from the symmetries or SB of equivalent atoms. Specifically, we can determine when the molecular orbitals preserve symmetries:

**Proposition 4.3.** *The global molecule feature learned by $\Phi_k^{\mathcal{M}}(\boldsymbol{x})$, as defined in equation 9, is equivariant if $c_{kj}(\boldsymbol{x}_j, a_j)$ are invariant and the atomic orbitals $\Phi_{(\boldsymbol{x}_j, a_j)}(\boldsymbol{x})$ are equivariant.*

## 5 SYMMETRY-ADAPTED NEURAL NETWORK

In this section, we present constitutions of our proposed symmetry-adapted neural network (SANN), a new GNN architecture inspired by the concept of SALCs. SANN is designed to achieve relaxed equivariance for learning spontaneous symmetry-broken features while compatible with the learning of invariant and equivariant features. This flexibility allows SANN to serve as a powerful tool for understanding molecular properties under varying symmetry constraints and SSB scenarios.

To effectively leverage molecular symmetries, SANN employs a new message-passing approach. This approach involves passing messages within and across sets of symmetry-equivalent atoms, respectively, followed by aggregation of information to capture the global features of molecules. In this section, we will introduce the key components of our model: the canonicalization and the construction of equivalence classes, the graph structures within and across equivalence classes, the design of symmetry-adapted message-passing mechanisms, and the loss function for learning SSB.

**Setup.** Let $\mathcal{M} = \{(\boldsymbol{x}_i, a_i)\}$ be a molecular data. In particular, each atom $a_i$ might be associated with a vector of features provided by the dataset, denoted by $\boldsymbol{a}_i$.

**Canonicalization and equivalence class construction.** SANN achieves the relaxed equivariance by leveraging canonicalization through a frame $\mathcal{F}$ (see Definition 3.5). Additionally, SANN relies on the construction of quotient set $\mathcal{M}/\operatorname{Sym}(\mathcal{M})$ from each molecular structure $\mathcal{M}$.

To simultaneously perform canonicalization and construct the quotient set, we adopt the asymmetric unit normalization (ASUN) framework proposed by Baker et al. (2024). Notably, the quotient set $\mathcal{M}/\operatorname{Sym}(\mathcal{M})$ and the asymmetric unit are closely related, with the latter being a minimal representative of the entire molecule selected from the former. Specifically, the asymmetric unit, $\operatorname{ASU}(\mathcal{M})$, is defined as the smallest subset of $\mathcal{M}$ such that applying the symmetry operations of $\operatorname{Sym}(\mathcal{M})$ to $\operatorname{ASU}(\mathcal{M})$ will recover the entire molecule $\mathcal{M}$, i.e. $\bigcup_{g \in \operatorname{Sym}(\mathcal{M})} g \cdot \operatorname{ASU}(\mathcal{M})$. It can be shown that $\operatorname{ASU}(\mathcal{M})$ is constructed by selecting a single representative element from each equivalence class in $\mathcal{M}/\operatorname{Sym}(\mathcal{M})$. In particular, we can express $\operatorname{ASU}(\mathcal{M})$ as a subset $\{(\boldsymbol{x}_{i_k}, a_{i_k})\} \subset \mathcal{M}$ where $[(\boldsymbol{x}_{i_k}, a_{i_k})] \neq [(\boldsymbol{x}_{i_{k'}}, a_{i_{k'}})]$ for any distinct $k, k'$. We report the details of our canonicalization in Appendix B.

**Graph structures within and across equivalence classes.** We use the elements within $\operatorname{ASU}(\mathcal{M})$ to build a graph structure across equivalence classes. For example, we can construct a graph using traditional radial cutoff or a fully connected graph on the set $\operatorname{ASU}(\mathcal{M}) = \{(\boldsymbol{x}_{i_{k'}}, a_{i_{k'}})\}$. Then we obtain a graph $\mathcal{G}_{\mathcal{M}/\operatorname{Sym}(\mathcal{M})} = (\mathcal{M}/\operatorname{Sym}(\mathcal{M}), \mathcal{E})$, where each node represents an equivalence class and two classes are connected if their representatives in ASU are connected by the graph construction. Similarly, for each set of equivalent atoms $[(\boldsymbol{x}_i, a_i)] \in \mathcal{M}/\operatorname{Sym}(\mathcal{M})$, we can construct a graph using radial cutoff or a fully connected on the set $[(\boldsymbol{x}_i, a_i)]$. We denote the resulting as $\mathcal{G}_{[(\boldsymbol{x}_i, a_i)]}$.

**Symmetry-adapted message-passing mechanism.** We denote the frame and the canonicalization of $\mathcal{M}$ constructed by ASUN as $\mathcal{F}(\mathcal{M}) \in \operatorname{O}(3)$ and $c(\mathcal{M}) := \{(\boldsymbol{x}_i^{\text{canonical}}, a_i)\}$, respectively, where $\boldsymbol{x}_i^{\text{canonical}}$ represents the coordinates of each atom after canonicalization. In particular, the initial feature $\boldsymbol{f}_i^0 = (\boldsymbol{x}_i^{\text{canonical}}, \boldsymbol{a}_i)$ is formed by concatenating the spatial feature $\boldsymbol{x}_i^{\text{canonical}}$ with atomic feature $\boldsymbol{a}_i$. This combined feature is then fed into the subsequent symmetry-adapted message-passing mechanism, which is defined as follows:

$$\boldsymbol{f}_{[(\boldsymbol{x}_i, a_i)]}^l = \operatorname{UPD}\left(\operatorname{AGG}\left(\{\boldsymbol{f}_i^l, \boldsymbol{f}_j^l \mid (i, j) \in \mathcal{G}_{[(\boldsymbol{x}_i, a_i)]}\}\right)\right)$$

$$\boldsymbol{g}^l = \operatorname{UPD}\left(\operatorname{AGG}\left(\{\boldsymbol{f}_{[(\boldsymbol{x}_i, a_i)]}^l, \boldsymbol{f}_{[(\boldsymbol{x}_j, a_j)]}^l \mid ([(\boldsymbol{x}_i, a_i)], [(\boldsymbol{x}_j, a_j)]) \in \mathcal{G}_{\mathcal{M}/\operatorname{Sym}(\mathcal{M})}\}\right)\right)$$

$$\boldsymbol{f}_i^{l+1} = \operatorname{Attention}(\boldsymbol{f}_i^l, \boldsymbol{f}_{[(\boldsymbol{x}_i, a_i)]}^l, \boldsymbol{g}^l)$$

where UPD and AGG are approximated by multilayer perceptrons (MLPs), $\boldsymbol{f}_{[(\boldsymbol{x}_i, a_i)]}^l$ denotes the feature learned within the equivalence class $[(\boldsymbol{x}_i, a_i)]$, and $\operatorname{Attention}$ denotes a self-attention MLP. Notably, the first equation defines message-passing within equivalence classes, the second defines message-passing across equivalence classes, and the third can be interpreted as an attention mechanism learning atomic features from the features within and across equivalence classes.

**Loss function.** To accurately measure the performance of our model on both symmetry-preserving (equivariant) and symmetry-breaking tasks, we employ the SBM defined in equation 3 with the

Chamfer distance (CD) – a faster proxy for the earth mover's distance (EMD) (Villani et al., 2009) – as our metric $m$. CD is a popular measure between point clouds for ML (Kusner et al., 2015; Wan et al., 2019; Bakshi et al., 2024). For point clouds $X, Y \in \mathbb{R}^n$, CD is defined as follows:

$$m(X, Y) = \sum_{x \in X} \min_{y \in Y} \|x - y\|_2^2.$$

CD is particularly effective for applications involving shape and spatial comparison. In contrast, mean absolute error (MAE) and mean squared error (MSE) operate on a strict element-wise correspondence, assuming perfect alignment between the compared sets. This limitation makes MAE and MSE less suitable for tasks where point order or arrangement may vary, and CD is a more robust measure of similarity in terms of structure and spatial proximity. We will refer to equation 3 using CD as **SBCD**.

## 6 EXPERIMENTAL RESULTS

We systematically test the efficacy of our approach. First, we demonstrate the theoretical limitations of equivariant neural networks and the advantages of our techniques. We then study the effects of the symmetry-breaking metric. Finally, we compare SANN to foundational invariant and equivariant GNNs on the benchmark QM7-X (Hoja et al., 2021) dataset. Through this sequence of studies, we observe that SANN makes significant advancements for symmetry-breaking tasks. We compare our method against other foundational methods. SchNet (Schütt et al., 2018), and DimeNet++ (Gasteiger et al., 2020) are chosen as invariant architectures, and EGNN (Satorras et al., 2021), and PaiNN Schütt et al. (2021) are chosen as and equivariant architectures. This provides a baseline for foundational architectures and state-of-the-art architectures for both invariant and equivariant models. Furthermore, to assess approximate equivariance we equip EGNN and PaiNN with noise injectivity into the positional data and label these models +Noise. We consider a noise parameter with a standard deviation between $0.001$ and $0.1$ and report the best results. We also enable symmetry breaking on all architectures using a two-layer multi-layer perceptron as an output layer and denote these architectures +MLP. Additional dataset, model, training, and hyperparameter optimization details are listed in Appendix C.

### 6.1 SPONTANEOUS SYMMETRY BREAKING

First we construct a dataset to determine the effects of incorporating SBM for disambiguation. The dataset is comprised of two identical square graphs with opposing orientations based on Fig. 3. The train, validate, and test sets are kept identical to eliminate generalizability as a limiting factor. Each model is trained for 100 epochs using the AdamW optimizer and a scheduled learning rate with a maximum value of $1e\text{-}4$. We train separate models with CD and SBCD loss functions. The results are reported in Table 1. The optimal distance without deforming the square is 0.25. We use this distance as a threshold to highlight models with smaller mean test loss over 10-fold cross validation.

**SBCD ablation study.** This task is an ablation study on SANN, which demonstrates that SBM is crucial for disambiguation. There are two areas where SSB limits model applicability. First, noisy or unclean datasets may contain re-occurring input with differing labels. This breaks the assumption of injectivity necessary for learning and leads to learning bottlenecks and inference errors. Second, clean data introduces training bias toward a particular symmetry-breaking orientation, significantly degrading the practical applicability of the model.

**Broken versus approximate equivariance.** This task also serves as a comparison between approximate equivariance, standard symmetry breaking, and the symmetry breaking measure. We observe that all models are incapable of learning without the equipped SBCD. Despite the ability of approximate equivariance to break symmetries, it cannot resolve SSB without SBCD.

|            | CD    | SBCD  |
|-----------:|-------|-------|
| SchNet     | 0.250 | 0.250 |
| DimeNet++  | 0.252 | 0.251 |
| EGNN       | 0.251 | 0.251 |
| PaiNN      | 1.766 | 1.776 |
| EGNN+Noise | 0.271 | 0.238 |
| PaiNN+Noise| 3.225 | 3.178 |
| SchNet+MLP | 3.877 | 3.921 |
| DimeNet+MLP| 1.911 | 2.000 |
| EGNN+MLP   | 0.748 | 0.381 |
| PaiNN+MLP  | 1.394 | 1.243 |
| SANN (Ours)| 0.250 | 0.132 |

Table 1: The mean test metric over 10-fold cross-validation on the SSB dataset. The minimum square to rectangle distance is 0.250, and highlighted results achieve lower loss.

## 6.2 Normal Modes of $H_2O$

To demonstrate the limitations of strict equivariance constraints, we construct a set of synthetic experiments. Molecules are in a state of constant motion, where the possible motions are called the *normal modes*. Figure 5 shows the normal modes of $H_2O$. The symmetries are shared by the symmetric stretching and bending motions in Fig. 5 (a) and (b), respectively. Although the asymmetric stretching motion in Fig. 5 (c) does not share any symmetry with the molecule, there is an inextricable relationship between the orientation of the motion and the molecular symmetries.

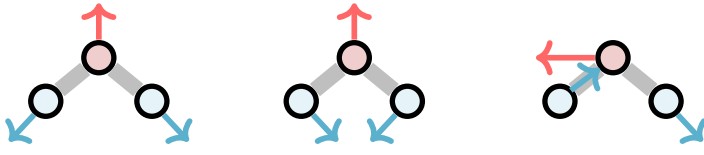

(a) Symmetric Stretch     (b) Bending Motion     (c) Asymmetric Stretch

Figure 5: The three normal modes of the $H_2O$ molecule. Arrows depict the motions, with oxygen and its motion in red, hydrogen in blue, and the bonds in gray. The molecular symmetries are shared by the (a) symmetric and (b) bending motion, but not by the (c) asymmetric stretching motions.

We design a set of synthetic tasks, one for each mode. Each dataset contains two $H_2O$ molecules, with the atomic numbers and position information as node features and featureless fully connected edges. The node labels are the directional vectors of the motions and differ by task. Finally, we apply a random rotation to each molecule and then center its point cloud at the origin. To assess a method's performance on a given motion, we train individual models on each task. The train, validate, and test sets are kept identical to eliminate generalizability as a limiting factor. Each model is trained for 100 epochs using the AdamW optimizer and a scheduled learning rate with a maximum value of $1e\text{-}4$. The loss and accuracy are measured in terms of SBCD. We perform 10-fold cross-validation on each task and report the results in Table 2. We cluster the results based on the architectures invariance (Invar.), equivariance (Equivar.), approximate equivariance (Appr. Equivar.) and symmetry breaking outputs. In invariant architectures, we predict a global invariant property and regress it with respect to the initial positions to make an equivariant prediction.

|  |  | | Symmetric | Bending Motion | Asymmetric |
|---|---|---|---|---|---|
| Invar. | | SchNet | $0.907 \pm 0.092$ | $0.924 \pm 0.101$ | $0.852 \pm 0.000$ |
| | | DimeNet | $0.954 \pm 0.095$ | $1.016 \pm 0.093$ | $0.863 \pm 0.009$ |
| Eqvar. | | EGNN | $0.000 \pm 0.000$ | $0.142 \pm 0.317$ | $0.852 \pm 0.000$ |
| | | PaiNN* | $0.011 \pm 0.010$ | $0.012 \pm 0.010$ | $0.008 \pm 0.009$ |
| Appr. Eqvar. | | EGNN+Noise | $0.002 \pm 0.000$ | $0.002 \pm 0.001$ | $0.806 \pm 0.027$ |
| | | PaiNN+Noise | $0.119 \pm 0.104$ | $0.087 \pm 0.068$ | $0.079 \pm 0.019$ |
| Symmetry Breaking | | SchNet+MLP | $0.801 \pm 0.144$ | $0.479 \pm 0.225$ | $0.571 \pm 0.290$ |
| | | DimeNet+MLP | $0.626 \pm 0.125$ | $0.134 \pm 0.123$ | $0.167 \pm 0.092$ |
| | | EGNN+MLP | $0.011 \pm 0.019$ | $0.002 \pm 0.004$ | $0.004 \pm 0.010$ |
| | | PaiNN+MLP | $0.006 \pm 0.004$ | $0.002 \pm 0.000$ | $0.008 \pm 0.009$ |
| | | SANN (Ours) | $0.003 \pm 0.002$ | $0.002 \pm 0.002$ | $0.041 \pm 0.079$ |

Table 2: Mean $\pm$ std. of the test SBCD for each model with 10-fold cross validation. Highlighting indicates a model's capability to learn consistently or its inability to learn at all: green shows that over all folds no SBCD is $\geq 10^{-1}$; red shows that over all folds the minimum SBCD$\geq 10^{-1}$.

Table 2 highlights our important findings. In particular, red highlighting denotes a model's incapacity to learn measured by the minimum SBCD over the 10-folds. We observe that invariant, approximate equivariant and symmetry breaking architectures are insufficient for producing accurate predictions. We also observe some unexpected results, namely that PaiNN performs well alone and without noise. Upon further investigation, we find that the vibrational modes can be recovered from the eigenvectors of the Hessian of the energy with respect to the position. This means that tensor-valued equivariant predictions are capable of learning normal modes. We have provided an

additional experiment in Table 7 of Appendix C empirically validating that PaiNN without higher order tensor predictions performs poorly.

### 6.3 QM7-X

To benchmark our results on real-world datasets, we consider a subset of the QM7-X dataset (Hoja et al., 2021), which contains many extended properties of small organic molecules, including the Hirshfeld dipole moments. These moments are symmetry-breaking but provide critical insight into inter-molecular charge transfer and polarization. A visualization of this SB is shown in Fig. 2. Our subset contains 225 molecules with five or fewer heavy atoms with details discussed in Table 3. Due to the small size of the dataset, we perform 10-fold cross-validation over random splits to account for generalizability. Table 3 reports the mean and standard

| Model | CD |
|---|---|
| SchNet | $0.0170 \pm 0.0078$ |
| SchNet+MLP | $0.749 \pm 0.454$ |
| EGNN | $0.0538 \pm 0.0149$ |
| EGNN+MLP | $0.0027 \pm 0.0007$ |
| SANN | $0.0012 \pm 0.00003$ |

Table 3: QM7-X Hoja et al. (2021) benchmark on 225 molecules, restricted to the set of conformers of a molecule with symmetry breaking. We split into 60/20/20% training, validation and testing 10-fold cross-validation of random splits.

deviation CD over the test data. We observe that SANN outperforms all other models. In addition, the force predictions for SchNet significantly beat the SchNet+MLP. However, we observe that the EGNN+MLP significantly outperforms the EGNN model, indicating some benefit from SB.

## 7    ADDITIONAL RELATED WORKS

We discuss some of the additional related works along two directions in the machine learning literature: relaxed equivariance and spontaneous symmetry breaking.

**Relaxed equivariance.** Relaxed equivariance has been studied in various papers. For instance, Kaba et al. (2023) introduce relaxed equivariance to ease the learning of invariant feature embeddings, which avoids using distinct homogeneous spaces in the framework of (Winter et al., 2022). Pertigkiozoglou et al. (2024) relax the hard equivariance constraint to ease the training of neural networks. These existing relaxed equivariance approaches typically involve relaxing constraints on specific layers of a neural network or learning canonicalization functions. This differs significantly from our proposed work, which leverages the canonicalization approach introduced in Baker et al. (2024) and avoids the need for additional learning processes. Moreover, these approaches are not applicable to learning symmetry-breaking. Besides relaxed equivariance, approximated equivariance has also been studied in various works; see, e.g., (Huang et al., 2024; Samudre et al., 2024).

**Symmetry breaking.** SB has been explored in various ML contexts, including physical systems (Wang et al., 2023b; Xie & Smidt, 2024; Lawrence et al., 2024) and the generation process of diffusion models; see, e.g., (Raya & Ambrogioni, 2023; Ambrogioni, 2023; Li & Chen, 2024). However, to the best of our knowledge, this paper is the first to investigate SB in ML-assisted molecular modeling. By applying canonicalization to address the ambiguity of SSB and developing a symmetry-adapted learning method, we effectively learn both symmetry-preserving and SB tasks simultaneously.

## 8    CONCLUSION

In this work, we have introduced SANN, a symmetry-adapted neural network for enhanced representation of molecular structures. SANN is capable of learning SSB via relaxed equivariance and our newly proposed SB measure. Our work highlights the existing limitations of symmetry-preserving neural networks using both synthetic tasks and real-world applications. Our work is limited by the scale of existing real-world molecular datasets with SB molecular properties, and the lack of accessible existing SB benchmarks. Our theoretical foundation for SB models pave the way for the construction of a multitude of SB architectures. We leave it to future work to further explore optimal SB mechanisms.

## 9 ETHICS STATEMENT

In this paper, we have proposed a new notion of relaxed equivariance informed by molecular inherent symmetries. The new symmetric-relaxation further motivates us to propose a new neural network model – named SLACNet. Our work aims to address some fundamental challenges in learning molecules and beyond, especially learning processes involving spontaneous symmetry breaking. Our work is purely methodological, and we validate our proposed approaches on the benchmark datasets. We do not expect to cause negative societal problems. To the best of our knowledge, we do not see any issues with potential conflicts of interest and sponsorship, discrimination/bias/fairness concerns, privacy and security issues, legal compliance, and research integrity issues (e.g., IRB, documentation, research ethics.

## 10 REPRODUCIBILITY STATEMENT

We are committed to conducting reproducible research, and we are achieving this through several avenues: (1) Comparing the novelty of our work against the literature. (2) Provide detailed derivations of the proposed approaches and theoretical results. (3) Conducting experiments using benchmark datasets. (3) Report experimental details. (4) Submitting the codes for all experiments with detailed documentation to ensure all experimental results are fully reproducible. All codes will be made publicly available.

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

# Appendix

## A MISSING PROOFS

*Proof of Proposition 3.6.* Recall that the canonicalization $f_{\text{canonical}}$ is defined as:

$$f_{\text{canonical}}(x) = \mathcal{F}(x) \cdot f(\mathcal{F}(x)^{-1} \cdot x).$$

Since $\mathcal{F}(x)^{-1} \cdot x$ is invariant, the relaxed equivariance of $f_{\text{canonical}}$ is directly inherited from the transformation the frame $\mathcal{F}(x)$. In particular, when $f$ is equivariant, we have

$$f_{\text{canonical}}(x) = \mathcal{F}(x) \cdot f(\mathcal{F}(x)^{-1} \cdot x) = \mathcal{F}(x) \cdot f_{\text{canonical}} = \mathcal{F}(x) f(\mathcal{F}(x)^{-1} \cdot x) \cdot f(x) = f(x)$$

This shows that $f_{\text{canonical}}(x)$ reduces to $f(x)$ when $f$ is equivariant. In other words, $f_{\text{canonical}}$ is equivariant. $\square$

*Proof of Proposition 4.1.* Suppose $g \in \text{Sym}(\mathcal{M})$. By the definition of equivalence classes, we have $g \cdot [(\boldsymbol{x}_i, a_i)] = [(\boldsymbol{x}_i, a_i)]$. This show that $\text{Sym}(\mathcal{M}) \subseteq \bigcap_{[(\boldsymbol{x}_i, a_i)] \in \mathcal{M}/\text{Sym}(\mathcal{M})} \text{Sym}([(\boldsymbol{x}_i, a_i)])$. Conversely, suppose $g \in \bigcap_{[(\boldsymbol{x}_i, a_i)] \in \mathcal{M}/\text{Sym}(\mathcal{M})} \text{Sym}([(\boldsymbol{x}_i, a_i)])$. Then for any $[(\boldsymbol{x}_i, a_i)] \in \mathcal{M}/\text{Sym}(\mathcal{M})$, we have $g \cdot [(\boldsymbol{x}_i, a_i)] = [(\boldsymbol{x}_i, a_i)]$. Since $\mathcal{M}$ is a disjoint union of all the equivalence classes in $\mathcal{M}/\text{Sym}(\mathcal{M})$, we see that $g \cdot \mathcal{M} = g \cdot \coprod_{[(\boldsymbol{x}_i, a_i)] \in \mathcal{M}/\text{Sym}(\mathcal{M})} [(\boldsymbol{x}_i, a_i)] = \coprod_{[(\boldsymbol{x}_i, a_i)] \in \mathcal{M}/\text{Sym}(\mathcal{M})} g \cdot [(\boldsymbol{x}_i, a_i)] = \coprod_{[(\boldsymbol{x}_i, a_i)] \in \mathcal{M}/\text{Sym}(\mathcal{M})} [(\boldsymbol{x}_i, a_i)] = \mathcal{M}$. This completes the proof. $\square$

*Proof of Corollary 4.2.* Let $g \in \text{Sym}(\mathcal{M})$ be a symmetry that is broken in $f(\mathcal{M})$. According to Proposition 4.1, we have

$$g \notin \text{Sym}(f(\mathcal{M})) = \bigcap_{[(\boldsymbol{x}_i, a_i)] \in f(\mathcal{M})/\text{Sym}(f(\mathcal{M}))} \text{Sym}([(\boldsymbol{x}_i, a_i)]).$$

This implies that $g$ does not lie in the point group of some equivalence class $[(\boldsymbol{x}_i, a_i)]$. Thus, the proof is complete. $\square$

*Proof of Proposition 4.3.* For any $g \in \text{O}(3)$, we need to show that $\Phi_k^{\mathcal{M}}(g^{-1}\boldsymbol{x}) = g \cdot \Phi_k^{\mathcal{M}}(\boldsymbol{x})$. Indeed,

$$
\begin{aligned}
\Phi_k^{\mathcal{M}}(g^{-1}\boldsymbol{x}) &= \sum_{\substack{(\boldsymbol{x}_j, a_j) \in [(\boldsymbol{x}_i, a_i)] \\ [(\boldsymbol{x}_i, a_i)] \in \mathcal{M}/\text{Sym}(\mathcal{M})}} c_{kj}(\boldsymbol{x}_j, a_j)\Phi_{(\boldsymbol{x}_j, a_j)}(g^{-1}\boldsymbol{x}) \\
&= \sum_{\substack{(\boldsymbol{x}_j, a_j) \in [(\boldsymbol{x}_i, a_i)] \\ [(\boldsymbol{x}_i, a_i)] \in \mathcal{M}/\text{Sym}(\mathcal{M})}} c_{kj}(\boldsymbol{x}_j, a_j)g \cdot \Phi_{(\boldsymbol{x}_j, a_j)}(\boldsymbol{x}) \\
&= g \cdot \sum_{\substack{(\boldsymbol{x}_j, a_j) \in [(\boldsymbol{x}_i, a_i)] \\ [(\boldsymbol{x}_i, a_i)] \in \mathcal{M}/\text{Sym}(\mathcal{M})}} c_{kj}(\boldsymbol{x}_j, a_j)\Phi_{(\boldsymbol{x}_j, a_j)}(\boldsymbol{x}) = g \cdot \Phi_k^{\mathcal{M}}(\boldsymbol{x}).
\end{aligned}
\tag{10}
$$

$\square$

## B ROBUST GRAPH AND FEATURE CONSTRUCTION

The reliable development of our architecture requires significant improvement of existing methods. We outline our adjustments in this section.

**Equivalence class construction.** The classical approach is to determine the point group of a molecule via guess-and-check and then construct the classes of symmetry-equivalent atoms. The minimal deterministic finite automata (DFA) (Hopcroft, 1971) constructed by the ASUN algorithm of Baker et al. (2024), exactly consists of the classes of symmetry-equivalent atoms. While ASUN provides a lightweight technique for handling noisy data, our theoretical results are reliant on a

robust construction of the equivalence classes. As a result, we enhance the tolerance mechanisms of ASUN for handling angular, co-linear, and co-planar symmetries. Our enhancements denoted ASUN+ achieve state-of-the-art results for canonicalizing QM9. Table 4 reports the mean earth movers distance (EMD) between the initial canonical representation and 16 folds for all molecular structures.

**Stabilizer construction.**  The point groups are nicely characterized in (Harris & Bertolucci, 1989), and an algorithm for point group detection is discussed. We follow this approach using the equivalence class construction from above to construct the stabilizers for the point cloud. We denote linear and planar structures as rank 1 and rank 2 respectively, following notation from (Baker et al., 2024). In Figure 6 we outline the algorithm for stabilizer construction. The starting point in each rank is to fix an axis of rotation. In rank 1 and rank 2 structures, this is already fixed by the nature of the data. In rank 3 it requires a slightly more expensive process of checking subsequently larger equivalence classes to determine if they have a mean which is not identically the origin. If they do then this point and the origin form an axis of possible rotation. Planar reflections that lie along the axis of rotation can be determined from the given axis of rotation.

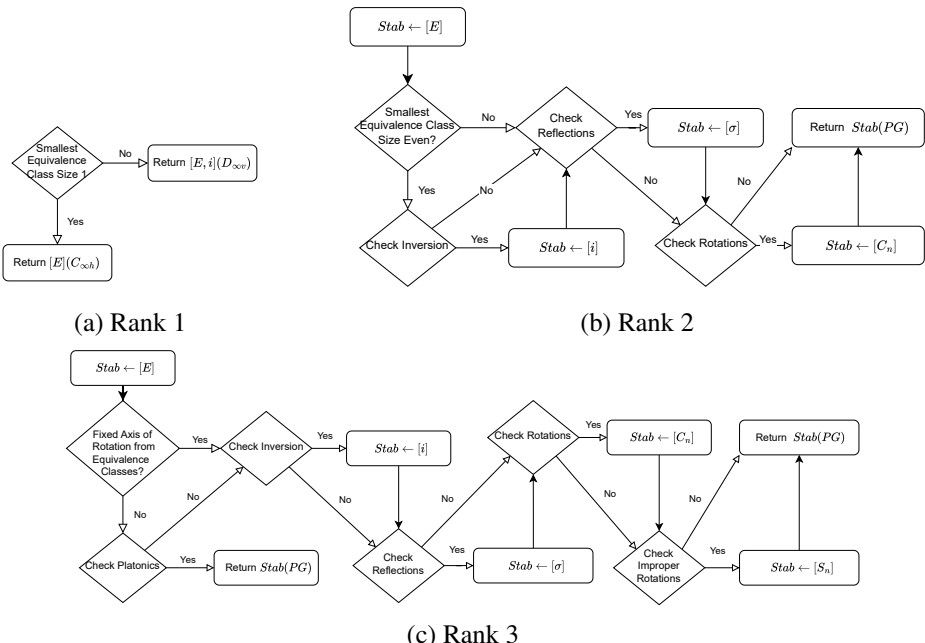

(a) Rank 1

(b) Rank 2

(c) Rank 3

Figure 6: Decision diagram for constructing the set of stabilizers from the equivalence class.

Notice that the set of molecular point groups which are infinite is $C_{\infty v}, D_{\infty h}, K_h$ Harris & Bertolucci (1989). $K_h$ represents perfect spherical symmetry which is not observed in practice. $C_{\infty v}, D_{\infty h}$ are linear molecules with or without a center of inversion, respectively. Because the data is aligned, the stabilizers are fixed to the z-axis and thus the relaxed metric only penalizes for discrepancies from the z-axis, and hence can be implemented as a finite summation.

**Graph Construction.**  For symmetric structures, the underlying graphs are computed by connecting the symmetry elements. Then select a set of representatives from the symmetry elements. The first representative is chosen without loss of generality. Each subsequent representative is selected from the subsequent set of symmetry elements by choosing the one with the minimum distance to the existing set of representatives. Again using the global symmetry of the molecule if there are multiple choices then the entire set of symmetry elements must be selectable and we choose without loss of generality. For non-symmetric structures, rather than constructing a fully connected graph, we use a radial cutoff with distance 8.

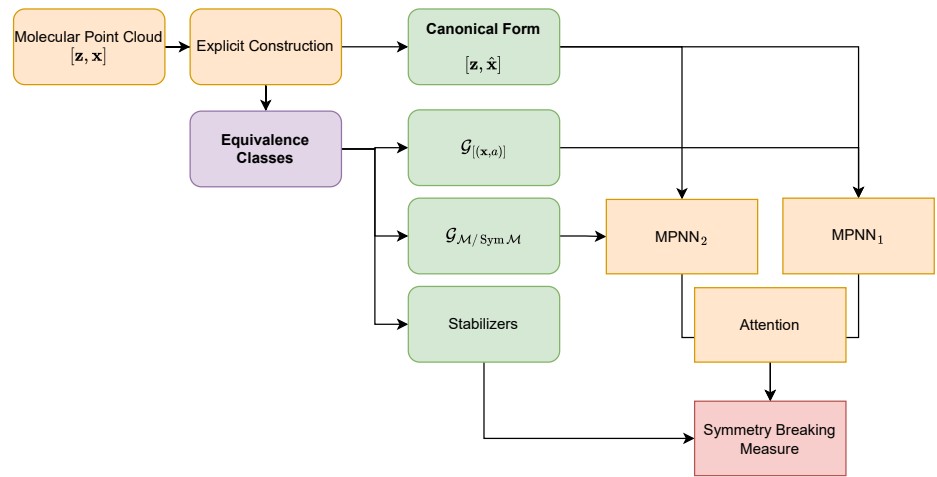

Figure 7: A diagram of the model architecture.

**Geometric feature construction.** We utilize the geometric feature construction from (Wang et al., 2022) to equip each edge with a set of geometric features. The Bessel basis function $\phi(x) = \frac{\sin(x)}{x}$ is not well-defined for when $x = 0$. The primary limitation here is that there can be no self-loops. However, our methodology requires self-loops in both graphs. We improve the numerical stability of this geometric feature construction by considering the Taylor series expansion about $x$. We also improve the numerical stability of the use of $\tan^{-1}(\frac{x}{y})$ by similar means.

## C  DATASETS AND METHODS

### C.1  MODEL ARCHITECTURE

In Figure 7 we provide a diagram of the model architecture and its key components for construction.

### C.2  QM7-X

The QM7-X data was curated to restrict to molecules and conformers where there is a lower state of symmetry between the Hirshfeld dipoles moment vectors and the initial structure. This is up to a tolerance parameter, which was selected to be large at .2 Angstroms. Because the QM7-X dataset contains multiple conformers, we have selected the conformers of these structures in addition to the structures with broken symmetry. In total, this is restricted to 255 molecular structures from the original dataset. To the best of our knowledge, there are no existing works applying graph neural networks to the entire dataset.

The QM7-X dataset contains small organic molecules with up to 7 heavy atoms and hydrogen. The dataset consists of 4 million molecules, with 42 molecular properties per molecule. For training on the QM7-X dataset, we follow the training procedures outlined by Satorras et al. (2021) for training EGNN on QM9. QM9 is a related dataset with up to 9 heavy atoms. Another key feature of QM7-X is that it contains duplicate molecules. While we do not specifically investigate the duplicate molecules in this work, we outline the advantage of our approach for handling duplicate molecules.

|  | Rank 1 | Rank 2 | Rank 3 |
|---|---|---|---|
| PCA | 0.00014 | 0.01793 | 0.82758 |
| AE | 1.15122 | 0.037539 | 0.03178 |
| ASUN | 0.00014 | 0.00008 | 0.02826 |
| **ASUN+** | **0.00000** | **0.00000** | **0.00009** |

Table 4: Mean (EMD) for QM9 canonicalization categorized by rank. Our adaptation ASUN+ achieves significantly improved results.

## C.3 HYPERPARAMETER OPTIMIZATION

For the foundational model hyperparameter optimization, we perform a grid search over a small set of hyperparameters. All other hyperparameters are taken from the best-reported model for the QM9 dataset.

| Hyperparameter | Options |
|---|---|
| Layers | $\{1, 2, 3, 5, 7\}$ |
| Hidden Features | $\{32, 64, 128\}$ |

Table 5: Hyperparameter tuning range for EGNN on synthetic tasks.

For SANN, we also perform a grid search over a small set of hyperparameters.

| Hyperparameter | Options |
|---|---|
| Layers | $\{1, 2, 3\}$ |
| Spherical Basis | $\{3, 4, 6\}$ |
| Radial Basis | $\{3, 4, 6\}$ |

Table 6: Hyperparameter tuning range for SANN on all tasks.

## C.4 INSUFFICIENCIES IN EXISTING DATASETS

The aim of this call for expanded datasets is to encourage additional datasets analyzing order-disorder and structural transitions Kivelson et al. (2024). In particular, we identify the cahllenges of incorporating two types of datasets, trajectory and interaction datasets.

**Trajectories.** Trajectory datasets like MD17 contain snapshots of structures during relaxation with the aim of predicting forces and energies from each snapshot. This equivariant task is marginal compared to predicting the relaxed structure from the initial structure, which requires symmetry breaking. Relaxed structure prediction is critical in drug design [4]. Initial trajectory datasets are under development [6] and we leave this to future work.

**Trajectories.** Trajectory datasets like MD17 contain snapshots of structures during relaxation with the aim of predicting forces and energies from each snapshot. This equivariant task is marginal compared to predicting the relaxed structure from the initial structure, which requires symmetry breaking. Relaxed structure prediction is critical in drug design [4]. Initial trajectory datasets are under development [6] and we leave this to future work.

**Interactions.** Interaction datasets, the most prominent being the open catalyst project, contains interactions between adsorbates and catalysts. The adsorbates are small molecules that are ideally suited to our architecture. However, the catalysts are crystalline structures with permutation group symmetries that are not directly addressed by our approach. Therefore, we leave this to future work.

## C.5 HIGHER-ORDER TENSORS FOR NORMAL MODES

We assess the ability of PaiNN to learn without the tensor-valued predictive output. The two types of models are denoted tensor-valued and vector-valued, respectively. The tensor-valued model utilizes the gated equivariant output. The vector-valued model predicts a global invariant feature and then regresses the value onto the original positions. We compare each type with +Noise and +MLP in Table 7. We observe that the tensor-valued PaiNN can learn the normal modes while the vector-valued PaiNN is not. This supports our conjecture that the normal modes, while ideal for symmetry breaking of vector-valued equivariant predictions, fail for higher-order representations.

| | | Symmetric | Bending Motion | Asymmetric |
|---|---|---|---|---|
| Tensor Valued | PaiNN | $0.011 \pm 0.010$ | $0.012 \pm 0.010$ | $0.008 \pm 0.009$ |
| | PaiNN+Noise | $0.119 \pm 0.104$ | $0.087 \pm 0.068$ | $0.079 \pm 0.019$ |
| | PaiNN+MLP | $0.006 \pm 0.004$ | $0.002 \pm 0.000$ | $0.008 \pm 0.009$ |
| Vector Valued | PaiNN | $3.470 \pm 1.300$ | $7.381 \pm 0.000$ | $11.19 \pm 14.51$ |
| | PaiNN+Noise | $6.782 \pm 0.920$ | $9.017 \pm 5.855$ | $22.13 \pm 16.86$ |
| | PaiNN+MLP | $6.240 \pm 2.260$ | $10.28 \pm 4.690$ | $4.826 \pm 1.170$ |

Table 7: Mean $\pm$ std. of the test SBCD for PaiNN variants with 10-fold cross validation. Highlighting indicates a model's capability to learn consistently or its inability to learn at all: green shows that over all folds no SBCD is $\geq 10^{-1}$; red shows that over all folds the minimum SBCD $\geq 10^{-1}$.

