# OpenReview forum: "Learning Molecular Symmetry Breaking via Symmetry-adapted Neural Networks"
_ICLR.cc/2025/Conference — Submitted to ICLR 2025_

### Official Review · Reviewer_SA1v · 2024-11-02

**Soundness:** 3
**Presentation:** 2
**Contribution:** 2
**Rating:** 5
**Confidence:** 4

**Summary:**

This paper proposes to examine the problem of symmetry breaking with equivariant neural networks in molecular regression tasks. For symmetric molecules, equivariance will force the output to have the symmetries as the input to the neural network, which sometimes limits the ability of a neural network to perform certain predictions. To tackle this problem, the authors propose to use adapted loss functions that are invariant to input stabilizers. They also suggest using relaxed equivariant prediction networks for this setting, which can be obtained through canonicalization. In practice, this is achieved with a canonicalization scheme based on asymmetric units. Experiments show that method used can break symmetries better than a combination of an equivariant neural network and an MLP and that the invariant loss helps significantly with learning.

**Strengths:**

- The work tackles an interesting problem with potentially important applications. The problem of symmetry breaking of equivariant neural networks has been discussed before, but not particularly its implications on molecular prediction tasks.
- The paper is mostly clear and easy to understand. The figures are also welcome and useful

**Weaknesses:**

**Major**
1. *Novelty and acknowledgment of previous work*
- I think there is generally an issue with this paper in terms of claiming novelty for things that were proposed in previous works and correctly acknowledging previous contributions.
- The most important case of this is the loss function Eq.3, which was already proposed and discussed by Xie and Smidt 2024. This is not acknowledged and is claimed to be an original contribution of the paper. The paper should be updated to not claim that this is an original contribution, and to correctly acknowledge Xie and Smidt 2024.
- The paper "Symmetry breaking and equivariant neural networks" (Kaba and Ravanbakhsh 2023) introducing relaxed equivariance as a solution to the symmetry breaking problem is also not mentioned although related. It should be mentioned and discussed.
- It also seems to me that there are instances in the paper where appropriate citations is not given to the right papers. For example, Baker et al. 2024 did not introduce either canonicalization or relaxed equivariance, yet the way in which the paper is cited seems to suggests that.
- I generally suggest to expand the related works section to discuss what solutions were previously proposed to the symmetry breaking problem and how the proposed one differentiates itself from these methods. For example, it is said that "[other] approaches are not applicable to learning symmetry breaking". What does that mean and how is that so, given the proposed method is based on canonicalization which is one of these other approaches?
2. *Incomplete experimental evaluation*
- I found that the baselines specifications was incomplete. I did not see explained in the paper what SchNet + MLP or EGNN + MLP means in terms of implementation. The fact that these models struggle on all tasks seems to indicate that they are simply not appropriate baselines. I suggest that the authors instead compare with Hofgard 2024 et al. or Lawrence et al. 2024 instead.
- An simple baseline that the should be compared against is simply symmetry breaking by adding noise to the inputs and then using an equivariant neural network.
3. *Regarding the method*
- The loss eq.3 seems of limited usefulness in practice. First it requires to find the stabilizer of all inputs, which may be costly. Second, and more importantly, its computation involves a potentially expensive optimization procedure especially if the stabilizer is large. Note that these limitations were already mentioned by Xie and Smidt 2024, I therefore find surprising that they are not at least discussed here. I think these limitations should be discussed as well as providing an analysis of computational cost.

**Minor**
- I did not find Figure 2 completely clear, what symmetry exactly is broken?
- The introduction mentions "crystal material melting" as an example of symmetry breaking. I think this is confusing/not accurate. Melting restores symmetry and does not break it. I suggest the authors precise this or use another example instead.
- Section 6.1 mentions "the Heisenberg uncertainty principle" as reason for why molecules are in constant motion and why the normal modes are of interest. I don't think that this is accurate, I suggest that the authors remove this reference.

**Questions:**

- I don't understand exactly why EMD was chosen as loss function and how it is computed.
- Could the authors explain how the QM7-X dataset was curated. What proportion of the data was retained? How does the method compare with baselines on the full dataset?
- As discussed in the paper and in previous work, canonicalization is sufficient to break symmetry. The additional clustering procedure based on equivalence classes is therefore not related to the symmetry breaking problem. I am therefore questioning the necessity of that design choice, what is the motivation? It is also necessary to perform ablation studies to understand the impact of that choice on result compared to simply using a message passing scheme.
- The term "non-functional" is used a few times? What is meant by that?
- I find the first paragraph of page 5 confusing. In particular, the statement "these outputs shared the same reduced symmetry" is not clear since the symmetry groups are not the same. They are conjugate, but that is the case for any objects in the same orbit. The fact that the rectangles in the example share the same symmetry is an accident, you could imagine a similar example with triangles where this would not be true anymore.

---

> ### Author Response · Authors · 2024-11-23
> **Response to Reviewer SA1v (part 1/3)**
>
> We thank the reviewer for the thoughtful review and valuable feedback. In what follows, we provide point-by-point responses to your comments.
>
> ----
>
> **W1: I think there is generally an issue with this paper in terms of claiming novelty for things that were proposed in previous works and correctly acknowledging previous contributions.**
>
> **Response**:  Thank you for pointing out the potential issues with acknowledging previous contributions. We kindly refer the reviewer to the first paragraph of Section 3, the second paragraph of Section 1 (top of page 2), and the final paragraph of Section 3.1 where we attributed credit to Kaba and Ravanbakhsh 2023, Xie and Smidt 2024, and Wang et al., 2024 among others. In our revised paper, we have incorporated a discussion of Hofgard 2024, and emphasized the novelty of our contributions in comparison to these papers.
>
> ----
>
> **W2: The most important case of this is the loss function Eq.3, which was already proposed and discussed by Xie and Smidt 2024. This is not acknowledged and is claimed to be an original contribution of the paper. The paper should be updated to not claim that this is an original contribution, and to correctly acknowledge Xie and Smidt 2024.**
>
> **Response**: We thank the reviewer for pointing out that the loss function has been proposed by Xie and Smidt 2024. We did not notice this loss function appeared in the appendix of the Xie and Smidt 2024. We have removed this from our contribution statement and have revised the related statement from the entire paper.
>
> ----
>
> **W3: The paper "Symmetry breaking and equivariant neural networks" (Kaba and Ravanbakhsh 2023) introducing relaxed equivariance as a solution to the symmetry breaking problem is also not mentioned although related. It should be mentioned and discussed.**
>
> **Response**: We kindly direct the reviewer to the beginning of Section 3, where we have outlined the contributions of Kaba and Ravanbakhsh (2023) to the concept of relaxed equivariance. To clarify further, while Kaba and Ravanbakhsh (2023) introduced the potential application of relaxed equivariance to the spontaneous symmetry breaking (SSB) problem, they did not develop specific methods or models to address it. In our work, we first provide a clear explanation of why relaxed equivariance is well-suited for SSB. We then propose a framework that leverages this concept to effectively learn SSB.
>
> ----
>
> **W4: It also seems to me that there are instances in the paper where appropriate citations is not given to the right papers. For example, Baker et al. 2024 did not introduce either canonicalization or relaxed equivariance, yet the way in which the paper is cited seems to suggests that.**
>
> **Response**: In our revised paper, we clarified the contributions of Baker et al. 2024, which provides an algorithm for canonicalization.
>
> ----
>
> **W5: I generally suggest to expand the related works section to discuss what solutions were previously proposed to the symmetry breaking problem and how the proposed one differentiates itself from these methods. For example, it is said that "[other] approaches are not applicable to learning symmetry breaking". What does that mean and how is that so, given the proposed method is based on canonicalization which is one of these other approaches?**
>
> **Response**:  Thank you for this suggestion. In our revised paper, we have both expanded the contributions section and added discussion emphasizing canonicalization, relaxed equivariance, and spontaneous symmetry breaking in the context of the related works.
>
> ----
>
> **W6: I found that the baselines specifications was incomplete. I did not see explained in the paper what SchNet + MLP or EGNN + MLP means in terms of implementation. The fact that these models struggle on all tasks seems to indicate that they are simply not appropriate baselines. I suggest that the authors instead compare with Hofgard 2024 et al. or Lawrence et al. 2024 instead.**
>
> **Response**: In SchNet+MLP and EGNN+MLP the output of each architecture has been fed through an additional two-layer MLP to break the symmetry enforcement. In the revision, we have emphasized this and provided a discussion that these architectures can perform well but do not always perform well as intended by the highlighting distinguishment.
>
> We notice that there is ambiguity because the reviewer did not leave precise citations for [1] and [2]. We have provided citations below and we hope the reviewer can provide some clarification.
>
> [1] Hofgard, Elyssa, Rui Wang, Robin Walters, and Tess Smidt. "Relaxed Equivariant Graph Neural Networks." arXiv, 30 July 2024, arXiv:2407.20471v1. https://arxiv.org/pdf/2407.20471.
>
> [2] Lawrence, Hannah, Vasco Portilheiro, Yan Zhang, and Sékou-Oumar Kaba. "Improving Equivariant Networks with Probabilistic Symmetry Breaking." OpenReview, 2024, https://openreview.net/pdf?id=1VlRaXNMWO.
>
> ----

---

> > ### Author Response · Authors · 2024-11-23
> > **Response to Reviewer SA1v (part 2/3)**
> >
> > **W7: An simple baseline that the should be compared against is simply symmetry breaking by adding noise to the inputs and then using an equivariant neural network.**
> >
> > **Response**: We appreciate your suggestion. This has been incorporated with a noise ablation study for spontaneous symmetry breaking in Section 6.1.
> >
> > ----
> >
> > **W8: The loss eq.3 seems of limited usefulness in practice. First, it requires to find the stabilizer of all inputs, which may be costly. Second, and more importantly, its computation involves a potentially expensive optimization procedure, especially if the stabilizer is large. Note that these limitations were already mentioned by Xie and Smidt 2024, I therefore find surprising that they are not at least discussed here. I think these limitations should be discussed as well as providing an analysis of computational cost.**
> >
> > **Response**: The stabilizers of the point groups are composed of $\(E\), \(C_n\),  \(i\)$, and/or $\(\sigma\)$ (reflection planes) [1]. Given sets of symmetry equivalent points, the stabilizer construction algorithm is similar to the point group detection algorithm [1].
> >
> > The algorithm follows a flow chart that systematically reduces the symmetries considered.
> > This is made even faster than standard point group detection by utilizing the sets of symmetry equivalent points. We provide a flow chart for our algorithm in Appendix B.
> >
> > [1] Harris, D. C., & Bertolucci, M. D. (1989). Symmetry and spectroscopy : an introduction to vibrational and electronic spectroscopy. Dover Publications.
> >
> > ----
> >
> > **W9: I did not find Figure 2 completely clear, what symmetry exactly is broken?**
> >
> > **Response**: In Figure 2 both molecular structures have a vertical mirror plane that contains the oxygen both carbon and two hydrogen atoms. On the left, the forces between the carbons and the oxygen also lie in this plane. On the right, the forces do not lie on this plane and hence break the vertical mirror symmetry provided by this plane. We have provided further clarification in the revised version.
> >
> > ----
> >
> > **W10: The introduction mentions "crystal material melting" as an example of symmetry breaking. I think this is confusing/not accurate. Melting restores symmetry and does not break it. I suggest the authors precise this or use another example instead.**
> >
> > **Response**: We acknowledge that melting is often associated with restoring symmetry. However, the specific example referenced in our manuscript discusses symmetry-breaking phenomena that can occur in certain crystal materials during melting processes. In this context, melting involves the disruption of a high-symmetry state, potentially leading to intermediate or lower-symmetry configurations under specific conditions. We have revised the related statement in the revised paper.
> >
> > ----
> >
> > **W11: Section 6.1 mentions "the Heisenberg uncertainty principle" as reason for why molecules are in constant motion and why the normal modes are of interest. I don't think that this is accurate, I suggest that the authors remove this reference.**
> >
> > **Response**: Thank you for pointing this out. This has been removed.
> >
> > ----
> >
> > **Q1: I don't understand exactly why EMD was chosen as loss function and how it is computed.**
> >
> > **Response**:  Thank you for bringing this to our attention. In practice, we use the Chamfer distance, which is often used as a proxy for the more computationally demanding EMD. The Chamfer distance measures the similarity between two point sets by averaging the squared distances from each point in one set to its nearest neighbor in the other.
> >
> > We have included a broader discussion about why this is a useful metric for comparing point clouds in the revision. It is a simple and efficient metric widely used in 3D shape analysis and computer vision.
> >
> > ----
> >
> > **Q2: Could the authors explain how the QM7-X dataset was curated. What proportion of the data was retained? How does the method compare with baselines on the full dataset?**
> >
> > **Response**: The QM7-X data was curated to restrict to molecules and conformers where there is a lower state of symmetry between the Hirshfeld dipoles moment vectors and the initial structure. This is up to a tolerance parameter, which was selected to be large at 0.2 Angstroms. Because the QM7-X dataset contains multiple conformers, we have selected the conformers of these structures in addition to the structures with broken symmetry. In total, this is restricted to 255 molecular structures from the original dataset. To the best of our knowledge, there are no existing works applying graph neural networks to the entire dataset. This is likely due to the size of the original dataset. We have incorporated this discussion into Appendix C.
> >
> > ----

---

> > > ### Author Response · Authors · 2024-11-23
> > > **Response to Reviewer SA1v (part 3/3)**
> > >
> > > **Q3**: As discussed in the paper and in previous work, canonicalization is sufficient to break symmetry. The additional clustering procedure based on equivalence classes is therefore not related to the symmetry breaking problem. I am therefore questioning the necessity of that design choice, what is the motivation? It is also necessary to perform ablation studies to understand the impact of that choice on result compared to simply using a message passing scheme.
> > >
> > > **Response**: Thank you for the feedback. We have incorporated an ablation study in Section 6.1.
> > >
> > > ----
> > >
> > > **Q4**: The term "non-functional" is used a few times? What is meant by that?
> > >
> > > **Response**: It means one input can correspond to multiple outputs, and we have made this point clear in the revision.
> > >
> > > ----
> > >
> > > **Q5**: I find the first paragraph of page 5 confusing. In particular, the statement "these outputs shared the same reduced symmetry" is not clear since the symmetry groups are not the same. They are conjugate, but that is the case for any objects in the same orbit. The fact that the rectangles in the example share the same symmetry is an accident, you could imagine a similar example with triangles where this would not be true anymore.
> > >
> > > **Response**: Thank you for highlighting this point. We have rephrased the statement to clarify its intended meaning. Specifically, the two rectangles differ only by a rotation, which results in their symmetry groups being related by conjugation under that rotation. When we stated that two objects "have the same symmetry," we were referring to the type of symmetry they exhibit, rather than their exact symmetry groups. This perspective emphasizes the qualitative nature of the symmetry rather than the specific group structure. To avoid confusion, we have revised the wording to say "the same type of symmetries" and clarified that it means the symmetry groups are related by conjugation.
> > >
> > >
> > > ------
> > >
> > > Thank you for considering our rebuttal. We appreciate your feedback and are happy to address further questions on our paper.

---

> ### Comment · Reviewer_SA1v · 2024-11-25
> **Response**
>
> I thank the authors for their answers. They have addressed some of my concerns, but not all.
>
> - Experiment 6.1:
>   - Is symmetry breaking required for this task? Why are the equivariant baselines performing well?
>   - The model with best performance should be bolded in Table 1 even if it's not the proposed method to be consistent with the rest of the paper.
>   - Ablation study regarding equivalence class scheme: I don't see a clear reference to this, what I see is an ablation study on the effect of the SBCD, which is a different point. I still don't see evidence that the clustering scheme based on equivalence classes is beneficial.
> - I find the description in 6.2 confusing. Here are my issues :
>   - It is not clear to me what exactly is predicted, is it the time evolution of the molecular configuration? That, and how is the prediction performed (what is given as input to the network) should be clarified. A sentence saying "in general, we use force regression to make predictions unless utilizing symmetry-breaking architectures." was added, but then what is used how are predictions made with symmetry-breaking architectures if not forces? Some clarification on this would be necessary.
>   - I don't why this task exemplifies the symmetry breaking problem. I have not fully understood what the task is, but I assume it is to predict the motion of the molecule. For that position, and velocity are necessary. Even in the asymmetric stretch configuration, the molecule doesn't have the same symmetry as the stretch, but the input to the neural network (positions and velocities) will not have special symmetries. So an equivariant network is expected to be able to perform this task. Maybe this is the reason why some equivariant baselines perform as well or better than the proposed method.
>   - The sentence "we see several expected results, in particular, that symmetry preserving and approximate symmetry methods are insufficient for resolving the broken symmetry." was added. This does not seem true, some equivariant perform fine and even outperform the proposed method, if I am not mistaken. The authors should be clear here and provide an explanation (see previous point). Assigning that to future work is not an option as this is related to the central point of the paper.
>   - I don't understand why the proposed method is bolded in Table 2 if some baselines perform as well or better. Again, for consistency, the best method for each task should be bolded.
> - The added experimental results have shown that either models relying on equivariant features (EGNN, PaiNN) or models with noise perform on par or better than the proposed method. In particular, some models are not supposed to solve the tasks properly due to equivariance and yet perform fine. The performance of the proposed method is therefore mitigated. Can the authors comment on that and suggest ways to update the paper accordingly?

---

> ### Author Response · Authors · 2024-11-28
> **Further Response to Reviewer SA1v (1/2)**
>
> We sincerely thank the reviewer for their thoughtful and constructive feedback and your engagement in the discussion of our work. We acknowledge that there were errors in some of our synthetic tasks, which we have now thoroughly verified and corrected. Based on your further feedback, we have revised the paper accordingly, with the changes highlighted in red.
>
> **Overview**:  Upon further review, we identified an issue where the labels were incorrectly centered, leading to zero values. This has been corrected to ensure the reported results are accurate. The results reaffirm our theoretical findings that SANN with SBCD is capable of achieving SSB. Additionally, we observed that EGNN+Noise is capable of SSB in certain cases. However, all other architectures fail to surpass the threshold of 0.25 in 10-fold cross-validation.
>
> We understand there is significant interest in identifying which architectures perform well on Task 6.2. To clarify, invariant architectures and approximate equivariant architectures do not perform well. In contrast, symmetry-breaking architectures perform well. Among the baseline models, PaiNN stands out as an outlier, performing well because the normal modes can be described by the eigenvalues of a higher-order tensor. Notably, introducing noise into PaiNN decreased its performance. As noted in the reviewer's earlier comments, symmetry-breaking combined with symmetry-breaking measures enables high performance. To address this further, we have added additional commentary and results to Appendix C.5 to support our conjecture regarding PaiNN.
>
> In what follows, we provide point-by-point responses to your further comments.
>
> -----
>
> **Q1: Is symmetry breaking required for this task (experiment 6.1)? Why are the equivariant baselines performing well?**
>
> **Response:** Task 6.1, upon further review, we discovered that the labels were incorrectly centered, resulting in the entire label becoming zero. We have corrected this issue and updated our reported values accordingly. The corrected results confirm our theoretical findings that SANN with SBCD is capable of achieving SSB. Furthermore, we observe that EGNN+Noise demonstrates SSB capability in certain cases. However, all other architectures fail to exceed the threshold of 0.25 in 10-fold cross-validation.
>
> -----
>
> **Q2: The model with best performance should be bolded in Table 1 even if it's not the proposed method to be consistent with the rest of the paper.**
>
> **Response:** We have removed the boldface for consistency with the results in Table 2.
>
> -----
>
> **Q3: Ablation study regarding equivalence class scheme: I don't see a clear reference to this, what I see is an ablation study on the effect of the SBCD, which is a different point. I still don't see evidence that the clustering scheme based on equivalence classes is beneficial.**
>
> **Response:** In Task 6.1, upon further review, we discovered that the labels were incorrectly centered, which caused the entire label to become zero. We have corrected this issue and updated our reported values accordingly. These results affirm our theoretical findings that SANN with SBCD is capable of achieving SSB. Additionally, we observe that EGNN+Noise demonstrates SSB capability in certain cases. However, all other architectures fail to exceed the threshold of 0.25 in 10-fold cross-validation. Notably, the results from this task underscore that our clustering mechanism is more effective at achieving SSB than simple noise injection.
>
>
> -----
>
> **Q4: It is not clear to me what exactly is predicted, is it the time evolution of the molecular configuration? That, and how is the prediction performed (what is given as input to the network) should be clarified. A sentence saying "in general, we use force regression to make predictions unless utilizing symmetry-breaking architectures." was added, but then what is used how are predictions made with symmetry-breaking architectures if not forces? Some clarification on this would be necessary.**
>
> **Response:** The initial position is provided, and the position at the endpoint of each indicated vector is predicted. A vector-valued equivariant feature can be predicted by taking the derivative of a global invariant feature, similar to regressing energies to predict forces. We leverage this principle to train on equivariant features. In symmetry-breaking architectures, instead of making a global invariant prediction, this layer is replaced with an MLP, and the model is trained on the output of the MLP.

---

> ### Author Response · Authors · 2024-11-28
> **Further Response to Reviewer SA1v (2/2)**
>
> **Q5: I don't know why this task exemplifies the symmetry-breaking problem. I have not fully understood what the task is, but I assume it is to predict the motion of the molecule. For that position and velocity are necessary. Even in the asymmetric stretch configuration, the molecule doesn't have the same symmetry as the stretch, but the input to the neural network (positions and velocities) will not have special symmetries. So an equivariant network is expected to be able to perform this task. Maybe this is the reason why some equivariant baselines perform as well or better than the proposed method.**
>
> **Response:** The input to the model is the indicated structure of the H$\_2$​O molecule, and the objective is to predict the structure where the positions align with the endpoints of each indicated vector. Among the baseline models, PaiNN stands out as an outlier, performing well due to its ability to describe the normal modes using the eigenvalues of a higher-order tensor. To further support our conjecture about PaiNN, we have included results without the tensor-valued output in Appendix C.5. This task effectively tests symmetry breaking in vector-valued equivariant predictions.
>
> -----
>
> **Q6: The sentence "we see several expected results, in particular, that symmetry preserving and approximate symmetry methods are insufficient for resolving the broken symmetry." was added. This does not seem true, some equivariant perform fine and even outperform the proposed method, if I am not mistaken. The authors should be clear here and provide an explanation (see previous point). Assigning that to future work is not an option as this is related to the central point of the paper.**
>
> **Response:** We have revised the text for clarity. The only outlier in this task is PaiNN, which provides both a tensor-valued equivariant output and a vector-valued output. This highlights a critical limitation of this synthetic task for tensor-valued equivariant architectures. To address this, we demonstrate directly that PaiNN, when restricted to exclude the tensor-valued output, is unable to learn effectively. The corresponding results have been included in Appendix C.5.
>
> -----
>
> **Q7: I don't understand why the proposed method is bolded in Table 2 if some baselines perform as well or better. Again, for consistency, the best method for each task should be bolded.**
>
> **Response:** Thank you for pointing this out. We previously used bold text to indicate our method but have removed it for consistency. Please refer to our responses above for detailed explanations of why certain methods perform well.
>
> -----
>
> **Q8: The added experimental results have shown that either models relying on equivariant features (EGNN, PaiNN) or models with noise perform on par or better than the proposed method. In particular, some models are not supposed to solve the tasks properly due to equivariance and yet perform fine. The performance of the proposed method is therefore mitigated. Can the authors comment on that and suggest ways to update the paper accordingly?**
>
> **Response:**  While several baselines equipped with symmetry-breaking sets demonstrate the ability to learn symmetry breaking, it is particularly noteworthy that our architecture can effectively handle equivariance despite not being inherently equivariant by design. We emphasize the importance of constructing the symmetry-breaking set and propose the SANN architecture as a compelling approach that warrants further exploration, especially in the context of architecture designs based on equivalence classes.
>
> -----
>
>
> Thank you once again for considering our rebuttal. Please do not hesitate to reach out if the response and revisions do not fully address your concerns. We sincerely appreciate it if you could let us know if all your concerns about our paper have been cleared.

---

> > ### Author Response · Authors · 2024-11-30
> > **Seeking for your further feedback**
> >
> > Dear Reviewer SA1v,
> >
> > We appreciate your effort in reviewing our paper and providing us with valuable feedback.
> >
> > Before the end of the discussion, would you mind letting us know if we have addressed your concerns on our paper?
> >
> > Thank you for your consideration.
> >
> >
> > Regards,
> >
> > Authors

---

> ### Comment · Reviewer_SA1v · 2024-11-30
> **Response**
>
> I acknowledge the authors' response, but some of my key concerns were not satisfyingly addressed.
> 1. The authors propose an architecture based on two components, first canonicalization and second "symmetry-adapted linear combinations". It was already known that canonicalization is sufficient to achieve symmetry breaking. The authors do not provide any (experimental) evidence that the symmetry adapted linear combinations help, as opposed to standard message passing. To assess that, an ablation study using a canonicalization scheme and a standard message passing GNN is expected.
> 2. Key details are missing from the description of experiment 6.2. What is the data generation process, how many samples were the networks trained on, what are the inputs and targets, etc. Basically, this experiment is far from being reproducible with the information provided in the paper. The same is true to a lesser extent with 6.1.
> 3. I am not satisfied with the explanation provided for the performance of PaiNN. PainNN is strictly equivariant and it was proved that equivariant networks cannot break symmetries. But PaiNN still performs well on this task. Two explanations are logically possible, either PaiNN is not strictly equivariant or the task does not require symmetry breaking. The explanation provided by the authors does not seem to fall in one of these categories and therefore does not convince me.
> 4. All the experiments are toy and it is not clear that the proposed method will be of interest in practical applications. Some experiments on a real dataset that is not hand-curated for the purpose of this paper are necessary.
>
> All in all, I think the paper would benefit from some more work on the experimental part, discussion of previous work and mention of limitations. I therefore maintain my score.

---

> > ### Author Response · Authors · 2024-11-30
> > **Thank you**
> >
> > Thank you for considering our rebuttal, and we appreciate your further feedback.

---

### Official Review · Reviewer_CPoo · 2024-11-04

**Soundness:** 3
**Presentation:** 4
**Contribution:** 3
**Rating:** 6
**Confidence:** 2

**Summary:**

It is worth noting that this paper lies primarily outside my area of research. Consequently, I will refrain from making strong judgments on its novelty and significance, as I am less familiar with this specific field and its literature.

This paper addresses spontaneous symmetry breaking (SBB) in AI-driven molecular modeling tasks. It introduces a novel loss function designed for SBB scenarios, which accounts for the inherent ambiguity in symmetry-breaking outcomes. To achieve relaxed equivariance, the paper proposes a specific canonicalization of the learnable model. Additionally, it presents a new message-passing framework, the Symmetry-Adapted Neural Network (SANN), that passes messages within and across sets of symmetry-equivalent atoms, respectively, followed by aggregation of information to capture the global features of molecules. Finally, several experiments on synthetic tasks are conducted, showing that SANN significantly outperforms the tested baselines.

Overall, I recommend acceptance and would consider raising my score even further if the authors successfully address my feedback.

**Strengths:**

* Well-written with a clear and logical structure
* Effective and informative figures and illustrations
* Provides solid background on the algebraic foundations relevant to this work
* Introduces an innovative message-passing framework

**Weaknesses:**

* Stronger baselines should be considered. For instance, many approximate equivariant models are now available and may perform significantly better on SBB tasks compared to strictly equivariant counterparts.
* Depending on the symmetry group, calculating the symmetry-breaking measure in Equation 3 might not be straightforward by simply evaluating the metric \( m(x,y) \) for each element in the group. How would this measure be computed in such cases? Or does this paper only consider finite groups? This needs to be clarified in the discussion.
* Although the paper is well-structured and provides essential foundational context, it lacks an explanation of why the proposed message-passing framework can address spontaneous symmetry breaking. This point requires further discussion in the paper.
* The figures illustrating symmetry-breaking concepts are excellent. An additional illustration that explains the proposed message-passing mechanism would also be very helpful.

Minor Issues
* Define the Earth Mover’s Distance, as it may be unfamiliar to some in the ML community.
* Further motivation for the proposed SANN architecture would be beneficial. For example, why is self-attention introduced only in the third part, rather than earlier in the message-passing process?

**Questions:**

How are the underlying graphs in the experiments constructed? Are they based on nearest neighbors, and what influence does the radius parameter have on performance? Clarification on this in the paper would be beneficial.

Additional questions are noted under the Weaknesses section.

---

> ### Author Response · Authors · 2024-11-23
> **Response to Reviewer CPoo (part 1/2)**
>
> We thank the reviewer for the thoughtful review and valuable feedback. In what follows, we provide point-by-point responses to your comments.
>
> ----
>
> **W1: Stronger baselines should be considered. For instance, many approximate equivariant models are now available and may perform significantly better on SBB tasks compared to strictly equivariant counterparts.**
>
> **Response**: We have included additional numerical results on testing state-of-the-art (SOTA) equivariant and approximate equivariant models in the revised paper. In particular, we have incorporated DimeNet++ and PaiNN as SOTA invariant and equivariant architectures, respectively. We incorporate approximate equivariant architectures by injecting noise into the positional data in the forward training pass as discussed in [1]. In the revised version, we have provided a study in Section 6.1. We observe that approximate equivariance is not capable of strict symmetry breaking.
>
> [1] Lawrence, Hannah, Vasco Portilheiro, Yan Zhang, and Sékou-Oumar Kaba. "Improving Equivariant Networks with Probabilistic Symmetry Breaking." OpenReview, 2024, https://openreview.net/pdf?id=1VlRaXNMWO.
>
> ----
>
> **W2: Depending on the symmetry group, calculating the symmetry-breaking measure in Equation 3 might not be straightforward by simply evaluating the metric (m(x,y) ) for each element in the group. How would this measure be computed in such cases? Or does this paper only consider finite groups? This needs to be clarified in the discussion.**
>
> **Response**: You are correct that the construction of the symmetry-breaking measure is only applicable to finite groups. Thank you for pointing this out. We have included terminology in the paper to reflect this and expanded the discussion of our algorithm for generating the stabilizers.
>
> Notice that the set of molecular point groups that are infinite is $\(C_{\infty v}, D_{\infty h}, K_h\)$ [1]. $\(K_h\)$ represents perfect spherical symmetry which is not observed in practice. $\(C_{\infty v}, D_{\infty h}\)$ are linear molecules with or without a center of inversion, respectively. Because the data is aligned, the stabilizers are fixed to the $z$-axis and thus the relaxed metric only penalizes for discrepancies from the $z$-axis, and hence can be implemented as a finite summation.
>
> [1] Harris, D. C., & Bertolucci, M. D. (1989). Symmetry and spectroscopy : an introduction to vibrational and electronic spectroscopy. Dover Publications.
>
> ----
>
> **W3: Although the paper is well-structured and provides essential foundational context, it lacks an explanation of why the proposed message-passing framework can address spontaneous symmetry breaking. This point requires further discussion in the paper.**
>
> **Response**: Thank you for your invaluable feedback. As outlined in Proposition 4.1, we demonstrate that the symmetry of the entire molecule is determined by the symmetries of the equivalence classes of its atoms. Building on this, Corollary 4.2 explains that when symmetry breaking (SB) occurs, it must manifest within at least one of these equivalence classes. We have included this explanation in the main text to provide the necessary context. Based on this insight, we propose a message-passing framework that first learns features at the level of equivalence classes, rather than directly from neighboring nodes in the graph. This approach effectively captures SB, as it is directly tied to SB within a class of equivalent atoms. We then aggregate the learned features across equivalence classes to retain the overall symmetry of the molecule, as inspired by the intersection described in Proposition 4.1. This framework enables a more targeted and efficient detection of SB within the molecular structure.
>
> ----
>
> **W4: The figures illustrating symmetry-breaking concepts are excellent. An additional illustration that explains the proposed message-passing mechanism would also be very helpful.**
>
> **Response**: Thank you for the suggestion. Due to the limited space for the illustration in the main text, we have incorporated it in Appendix C.1 of the revised version.
>
> ----
>
> **W5: Define the Earth Mover’s Distance, as it may be unfamiliar to some in the ML community.**
>
> **Response**: Thank you for bringing this to our attention. In practice, we use the Chamfer distance, which is often used as a proxy for the more computationally demanding EMD. The Chamfer distance measures the similarity between two point sets by averaging the squared distances from each point in one set to its nearest neighbor in the other.
>
> We have included a broader discussion about why this is a useful metric for comparing point clouds in the revision. It is a simple and efficient metric widely used in 3D shape analysis and computer vision.
>
> ----

---

> > ### Author Response · Authors · 2024-11-23
> > **Response to Reviewer CPoo (part 2/2)**
> >
> > **W6: Further motivation for the proposed SANN architecture would be beneficial. For example, why is self-attention introduced only in the third part, rather than earlier in the message-passing process?**
> >
> > **Response**: We aim to leverage hierarchical information. Self-attention is critical for merging the information from the hierarchical architecture. It is worth to trying to incorporate attention in the first two but we did not try this.
> >
> > ----
> >
> > **Q1: How are the underlying graphs in the experiments constructed? Are they based on nearest neighbors, and what influence does the radius parameter have on performance? Clarification on this in the paper would be beneficial.**
> >
> > **Response**:
> > ***Symmetric Structure:*** The underlying graphs are computed by connecting the symmetry elements. Then select a set of representatives from the symmetry elements. The first representative is chosen without loss of generality. Each subsequent representative is selected from the subsequent set of symmetry elements by choosing the one with the minimum distance to the existing set of representatives. Again using the global symmetry of the molecule if there are multiple choices then the entire set of symmetry elements must be selectable and we choose without loss of generality.
> >
> > ***Non-symmetric Structure:*** Rather than constructing a fully connected graph, we use a radial cutoff with distance 8.
> >
> > We have provided a detailed description of this selection in Appendix B.
> >
> >
> > ------
> >
> > Thank you for considering our rebuttal. We appreciate your feedback and are happy to address further questions on our paper.

---

> ### Comment · Area_Chair_YfWD · 2024-11-27
> **Rebuttal Response Requested**
>
> Dear Reviewer,
> Do you mind letting the authors know if their rebuttal has addressed your concerns and questions? Thanks!
> -AC

---

> > ### Comment · Reviewer_CPoo · 2024-12-01
> >
> > I thank the authors for their answers. While I remain in favor of accepting the paper and I am overall happy with the author's replies, I do agree with the criticisms of my fellow reviewers regarding some experimental shortcomings (e.g. Reviewer SA1v). Thus I keep my score of marginally above the acceptance threshold.

---

> > > ### Author Response · Authors · 2024-12-02
> > > **Thank you**
> > >
> > > Thank you for considering our rebuttal and we appreciate your feedback.

---

### Official Review · Reviewer_ZVP2 · 2024-11-04

**Soundness:** 2
**Presentation:** 3
**Contribution:** 2
**Rating:** 5
**Confidence:** 3

**Summary:**

The paper proposes a method to relax equivariance constraints in molecules by utilizing a canonicalization technique and a modified loss function. This approach targets scenarios where symmetry breaking occurs, specifically based on the Curie principle.

**Strengths:**

The paper effectively shows the advantages of its approach using synthetic tasks and QM7-X dataset, illustrating improvements over equivariant models.

**Weaknesses:**

- The paper lacks evaluations and comparisons to a broader range of baseline models. While it compares its results against EGNN and SchNet, multiple equivariant models in the literature are not considered.

- The experimental setup and model evaluation could be expanded to include more benchmarks beyond QM7-X to fully assess the generalizability of the proposed approach.

**Questions:**

The authors claim this is the first work that tackles symmetry breaking in molecules. I think this needs to be revised, considering some works in the literature (1, 2, 3), and the paper already cited on crystals (4).

1. Presilla, Carlo, and Jona-Lasinio, Giovanni. "Spontaneous symmetry breaking and inversion-line spectroscopy in gas mixtures." Physical Review A, vol. 91, no. 2, Feb. 2015

2. Goryachev, A.B., and Leda, M. "Many roads to symmetry breaking: molecular mechanisms and theoretical models of yeast cell polarity." Molecular Biology of the Cell, vol. 28, no. 3, Feb. 2017

3. Langer, Marcel F., Pozdnyakov, Sergey N., and Ceriotti, Michele. "Probing the effects of broken symmetries in machine learning." 2024

4. Wang, Rui, Hofgard, Elyssa, Gao, Han, Walters, Robin, and Smidt, Tess E. "Discovering Symmetry Breaking in Physical Systems with Relaxed Group Convolution." 2024

---

> ### Author Response · Authors · 2024-11-23
> **Response to Reviewer ZVP2 (part 1/2)**
>
> We thank the reviewer for the thoughtful review and valuable feedback. In what follows, we provide point-by-point responses to your comments.
>
> ----
>
> **W1: The paper lacks evaluations and comparisons to a broader range of baseline models. While it compares its results against EGNN and SchNet, multiple equivariant models in the literature are not considered.**
>
> **Response**: In the revised paper, we have expanded the comparative analysis to include additional state-of-the-art models. Specifically, we have incorporated DimeNet++ and PaiNN for each task, broadening the range of invariant and equivariant architectures evaluated. We believe this addition provides a more comprehensive comparison and addresses your concerns regarding the scope of baseline models.
>
> -----
>
> **W2: The experimental setup and model evaluation could be expanded to include more benchmarks beyond QM7-X to fully assess the generalizability of the proposed approach.**
>
> **Response**: We have further studied approximate equivariant architectures by injecting noise into the positional data during training following paper [1]. We summarize the limitations of existing datasets below.
>
> ***Properties:*** The Hirshfeld dipole moment represents the charge separation within the molecule or material, offering a more nuanced understanding of molecular polarization. The Hirschfeld approach has been used to design semiconductors and dielectric materials where precise control of polarization is necessary [2]. In lead optimization, Hirshfeld dipole analysis has identified regions of polarity that need adjustment to enhance binding affinity or solubility [3]. For this reason, we feel that our property prediction is sufficient to assess the proposed approach.
>
> ***Trajectories:*** MD17 contains snapshots of structures during relaxation to predict forces and energies from each snapshot. This equivariant task is marginal compared to predicting the relaxed structure from the initial structure, which requires symmetry breaking. Relaxed structure prediction is critical in drug design [4].
>
> ***Interactions:*** The open catalyst project contains interactions between adsorbates and catalysts. The adsorbates are small molecules that are ideally suited to our architecture. However, the catalysts are crystalline structures with permutation group symmetries that are not directly addressed by our approach. Therefore, we leave this to future work.
>
> [1] Lawrence, Hannah, Vasco Portilheiro, Yan Zhang, and Sékou-Oumar Kaba. "Improving Equivariant Networks with Probabilistic Symmetry Breaking." OpenReview, 2024, https://openreview.net/pdf?id=1VlRaXNMWO.
>
> [2] Spackman, Mark A., and Patrick G. Byrom. "A Novel Definition of a Molecule in a Crystal." Chemical Physics Letters, vol. 267, no. 3–4, 1997, pp. 215–220. ScienceDirect, https://doi.org/10.1016/S0009-2614(97)00100-0.
>
> [3] Bultinck, Patrick, et al. "Critical Analysis and Extension of the Hirshfeld Atoms in Molecules." Journal of Computational Chemistry, vol. 23, no. 1, 2002, pp. 822-834. Wiley Online Library, https://onlinelibrary.wiley.com/doi/full/10.1002/jcc.10241.
>
> [4] De Vivo, Marco, Matteo Masetti, Giovanni Bottegoni, and Andrea Cavalli. "Role of Molecular Dynamics and Related Methods in Drug Discovery." Journal of Medicinal Chemistry, vol. 59, no. 9, 2016, pp. 4035-4061. American Chemical Society, https://doi.org/10.1021/acs.jmedchem.5b01684.
>
> ---

---

> > ### Author Response · Authors · 2024-11-23
> > **Response to Reviewer ZVP2 (part 2/2)**
> >
> > **Q1: The authors claim this is the first work that tackles symmetry breaking in molecules. I think this needs to be revised, considering some works in the literature (1, 2, 3), and the paper already cited on crystals (4).**
> >
> > **Response**: In our revised paper, we have incorporated a discussion of [1,2,3] and emphasized the novelty of our contributions in comparison to these papers. In particular our direct construction of the stabilizer set for the point groups. Each of these papers is a targeted application.
> >
> > References [1] and [2] target applications of spontaneous symmetry-breaking (SSB) to specific cases, such as chiral or prechiral molecules (like NH₃ and ND₃) [1] and cellular polarity [2]. Both establish methods for handling particular SSB but do not construct a general framework for point groups. Paper [3] utilizes random rotational augmentation which is a form of approximate equivariance for symmetry breaking. In the revised version, we have incorporated random data perturbations to study approximate equivariance. And provided a study in Section 6.1. We observe that approximate equivariance is not capable of strict symmetry breaking.
> >
> > [1] Presilla, Carlo, and Jona-Lasinio, Giovanni. "Spontaneous symmetry breaking and inversion-line spectroscopy in gas mixtures." Physical Review A, vol. 91, no. 2, Feb. 2015
> >
> > [2] Goryachev, A.B., and Leda, M. "Many roads to symmetry breaking: molecular mechanisms and theoretical models of yeast cell polarity." Molecular Biology of the Cell, vol. 28, no. 3, Feb. 2017
> >
> > [3] Langer, Marcel F., Pozdnyakov, Sergey N., and Ceriotti, Michele. "Probing the effects of broken symmetries in machine learning." 2024
> >
> > ------
> >
> > Thank you for considering our rebuttal. We appreciate your feedback and are happy to address further questions on our paper.

---

> > > ### Comment · Reviewer_ZVP2 · 2024-11-27
> > >
> > > I thank the authors for answering my questions. However, I'll keep my score, but similar to other reviewers, I still have some concerns about the evaluation of the proposed method. Additional small comment: I'm not sure about the sentence 'the first explicit construction of equivariant neural networks that enable learning SB leveraging molecular inherent symmetries' after the authors pointed out the previous works in symmetry breaking.

---

> ### Author Response · Authors · 2024-11-28
> **Further Response to Reviewer ZVP2**
>
> We sincerely thank the reviewer for their engagement in the discussion of our work. We acknowledge that there were small errors in some of our synthetic tasks (see details in our further response to Reviewer SA1v), which we have now thoroughly verified and corrected. Based on further reviewer feedback, we have revised the paper accordingly, with the changes highlighted in red.
>
> **Response**: We have revised this sentence to emphasize the novelty of our approach.
>
> We hope this addresses the reviewer's concerns. We would greatly appreciate it if the reviewer could provide more specific details regarding any remaining concerns with our paper. This will help us address them more effectively and ensure that our revisions meet the expectations of the review.

---

> > ### Author Response · Authors · 2024-11-30
> > **Seeking for your further feedback**
> >
> > Dear Reviewer ZVP2
> >
> > We appreciate your effort in reviewing our paper and providing us with valuable feedback.
> >
> > Before the end of the discussion, would you mind letting us know if we have addressed your concerns on our paper?
> >
> > Thank you for your consideration.
> >
> >
> > Regards,
> >
> > Authors

---

### Official Review · Reviewer_d1hA · 2024-11-04

**Soundness:** 3
**Presentation:** 3
**Contribution:** 2
**Rating:** 5
**Confidence:** 3

**Summary:**

The paper introduces a method to handle molecular modeling tasks that involve symmetry breaking—a phenomenon where a system transitions from a state with high symmetry to one with lower symmetry. Traditional equivariant models may struggle with spontaneous symmetry breaking (SSB) due to the constraints of equivariance. The proposed model decomposes molecules into symmetry-equivalent atom classes and performs message-passing within and across these classes. The paper also proposes a new symmetry-breaking measure (SBM) as a loss function to account for ambiguous outcomes in SSB. The author evaluate their model on synthetic tasks and molecular benchmark QM7-X.

**Strengths:**

The paper’s theoretical framework for spontaneous symmetry breaking (SSB), relaxed equivariance, and the proposed symmetry-breaking measure (SBM) is well-developed and accessible.

**Weaknesses:**

The significance of symmetry breaking in related fields, while highlighted, remains somewhat unclear. Both the number and scale of real-world experiments are limited. The authors attribute this to the lack of accessible datasets, but further evidence demonstrating the importance of symmetry breaking in relevant domains would strengthen the argument that this is not merely an edge case.

**Questions:**

Could the authors kindly provide more concrete evidence where symmetry breaking plays a critical role in molecular modeling tasks?

---

> ### Author Response · Authors · 2024-11-22
> **Response to Reviewer d1hA**
>
> We thank the reviewer for the thoughtful review and valuable feedback. In what follows, we provide point-by-point responses to your comments.
>
> ----
>
> **Q1: The significance of symmetry breaking in related fields, while highlighted, remains somewhat unclear. Both the number and scale of real-world experiments are limited. The authors attribute this to the lack of accessible datasets, but further evidence demonstrating the importance of symmetry breaking in relevant domains would strengthen the argument that this is not merely an edge case. Could the authors kindly provide more concrete evidence where symmetry breaking plays a critical role in molecular modeling tasks?**
>
>
> **Response**:  We appreciate the reviewer’s invaluable suggestions. We have added a discussion in Appendix C.4 to stress the importance of learning symmetry-breaking while ensuring the model’s equivariance guarantees. We summarize the crucial points in the following:
>
> - For a foundational reference on symmetry breaking in the mechanics of molecular structures, we refer the reviewer to [1]. This text covers **order-disorder and structural transitions** as well as the study of **crystalline solids, liquid crystals, ferromagnetism, superfluids, and superconductors**.
>
> - Properties: The **Hirshfeld dipole moment** represents the charge separation within the molecule or material, offering a more nuanced understanding of molecular polarization. The Hirschfeld approach has been used to design semiconductors and dielectric materials where precise control of polarization is necessary [2]. In lead optimization, Hirshfeld dipole analysis has identified regions of polarity that need adjustment to enhance binding affinity or solubility [3].
>
> - Trajectories: MD simulations introduce asymmetry by capturing **thermal fluctuations** as structures **minimize energy** over time. These processes perturb the symmetric starting structure. Understanding these perturbations is critical to drug design [4]. The MD17 dataset contains snapshots of structures during relaxation to predict forces and energies from each snapshot. This equivariant task is marginal compared to predicting the relaxed structure from the initial structure, which requires symmetry breaking.
>
> - Interactions: The open catalyst project contains interactions between adsorbates and catalysts. The adsorbates are small molecules that are ideally suited to our architecture. However, the catalysts are crystalline structures with permutation group symmetries that are not directly addressed by our approach. Therefore, we leave this to future work.
>
> [1] Kivelson, S. A., J. M. Jiang, and J. Chang. Statistical Mechanics of Phases and Phase Transitions. Princeton University Press, 2024. Google Books.
>
> [2] Spackman, Mark A., and Patrick G. Byrom. "A Novel Definition of a Molecule in a Crystal." Chemical Physics Letters, vol. 267, no. 3–4, 1997, pp. 215–220. ScienceDirect, https://doi.org/10.1016/S0009-2614(97)00100-0.
>
> [3] Bultinck, Patrick, et al. "Critical Analysis and Extension of the Hirshfeld Atoms in Molecules." Journal of Computational Chemistry, vol. 23, no. 1, 2002, pp. 822-834. Wiley Online Library, https://onlinelibrary.wiley.com/doi/full/10.1002/jcc.10241.
>
> [4] De Vivo, Marco, Matteo Masetti, Giovanni Bottegoni, and Andrea Cavalli. "Role of Molecular Dynamics and Related Methods in Drug Discovery." Journal of Medicinal Chemistry, vol. 59, no. 9, 2016, pp. 4035-4061. American Chemical Society, https://doi.org/10.1021/acs.jmedchem.5b01684.
>
> ------
>
> Thank you for considering our rebuttal. We appreciate your feedback and are happy to address further questions on our paper.

---

> ### Comment · Area_Chair_YfWD · 2024-11-27
> **Rebuttal Response Requested**
>
> Dear Reviewer,
> Do you mind letting the authors know if their rebuttal has addressed your concerns and questions? Thanks!
> -AC

---

> ### Comment · Reviewer_d1hA · 2024-12-02
>
> Thanks for the response. However, it does not fully address my concerns regarding the significance of symmetry breaking and the overall contribution of the work. Convincing and concrete evidence of the importance of symmetry breaking and the paper's contributions would be demonstrated by improved performance on real-world tasks where most current state-of-the-art models struggle. However, the experimental evaluation on the real-world dataset, QM7-X, includes only Schnet and EGNN as baselines, omitting comparisons with many recent state-of-the-art equivariant models. This significantly weakens the empirical support for the contribution.
>
> In addition, the weakness highlighted by my fellow reviewers raised my concern regarding the novelty of the work. E.g., the authors claimed that "we propose a new symmetry breaking measure (SBM)" and even listed this as the first point of the contribution in their initial submission. But Reviewer SA1v pointed out that this has already been proposed by previous work and wasn’t acknowledged. While this was corrected in the revised version, the novelty of the work is thus limited.
>
> I’m unable to raise my rating given these issues.

---

### Author Response · Authors · 2024-11-22
**General Response**

We thank the reviewers for their thoughtful review and valuable feedback, which have helped us significantly improve the paper. In response to the reviewers' suggestions, we have included additional experiments, clarified a few key points - especially credited to the original work that proposes the symmetry-breaking measure, and incorporated additional references in the revised paper. All revisions have been highlighted in blue in the revised paper.

A common comment from reviewers lies in insufficient experimental results to confirm the efficacy of the proposed approach and the importance of learning symmetry-breaking. As such, we have included additional numerical results on testing state-of-the-art (SOTA) equivariant and approximate equivariant models in the revised paper. In particular, we have incorporated DimeNet++ and PaiNN as SOTA invariant and equivariant architectures, respectively. Moreover, we have further studied approximate equivariant architectures by injecting noise into the positional data during training following paper [1].


Furthermore, we have made the following major revisions to the revised paper:

- We have credited Xie and Smidt, 2024 for introducing the symmetry-breaking measure.


- We have emphasized our **deterministic method** for constructing the set of stabilizers for a general point cloud.


- We have emphasized SANN architecture as a **new architecture for handling varying symmetry conditions** including invariant, equivariant, and symmetry-broken features.


[1] Lawrence, Hannah, Vasco Portilheiro, Yan Zhang, and Sékou-Oumar Kaba. "Improving Equivariant Networks with Probabilistic Symmetry Breaking." OpenReview, 2024, https://openreview.net/pdf?id=1VlRaXNMWO.



-----

We are glad to answer your further questions on our submission.

---

### Author Response · Authors · 2024-11-25
**Follow up**

Dear Reviewers,

Thank you for your effort in reviewing our paper and providing invaluable feedback.

We have done our best to provide thorough responses to each comment provided by the reviewers in the rebuttal. As the discussion period is ending soon, we kindly ask if you could review our responses to your review. If you have further questions or comments, we will do our best to address them before the discussion period ends.

Thank you for your consideration, and we appreciate your effort in reviewing our paper and considering our rebuttal.



Regards,

Authors

---

### Meta-Review · Area_Chair_YfWD · 2024-12-20

**Metareview:**

**Summary** This paper address the problem of spontaneous symmetry breaking in molecular modeling in which a system may transition from a high symmetry state to a lower symmetry state. While strictly equivariant models cannot map from the high symmetry state to the low symmetry state, relaxed equivariant models can.   The proposed method, Symmetry-Adapted Neural Networks (SANN) uses canonicalization and message passing across and within the equivalence classes of atoms in a molecule to achieve relaxed equivariance.

**Strength** This paper provides an interesting theoretical framework for building and evaluating models which can accommodate spontaneous symmetry breaking.  Reviewers generally agreed the paper was well written and praised the figures in particular.  Although there are problems with the empirical evaluations, the experiments of QM7-X do show the method can help.  Overall the method is a novel strategy to incorporate symmetry breaking into model and addresses a problem with potentially important applications.

**Weaknesses**  Reviewers agreed broadly that the experimental evaluation was lacking.  A common concern was lack of strong baselines including other SoTA equivariant architectures and other recents works addressing symmetry breaking. Although the authors have added some new baselines during the rebuttal, the reviewers still found the experiments lacking. Other issues include the need for ablations, additional details and analysis of the results, and more real-world tasks. A potential limitation of the approach is the difficulty of computing the symmetry breaking measure. Lastly, reviewers pointed out some claimed contributions such as the symmetry breaking measure had appeared in prior work which the authors addressed in their rebuttal.

**Conclusion** It is broadly the consensus of the AC and reviewers that although the method is promising the experiments require additional work before publication.

**Additional Comments On Reviewer Discussion:**

d1hA who gave a score 5, declined to increase their score due to lack of experiments on real world data beating SOTA equivariant baselines and due to lack of novelty due to the symmetry breaking measure being previously proposed.   ZVP2 also kept their score at 5 over similar concerns; the authors addition of 2 baselines did not adequately address their concern on evaluation.  SA1v pointed out some contributions were not new and the authors correctly attributed them.  SA1v also suggest new baselines which the authors added.

---

### Decision · Program_Chairs · 2025-01-22

Reject